# Modeled variations of the inherent optical properties of summer Arctic ice and their effects on the radiation budget: A case based on ice cores from CHINARE 2008–2016

Miao Yu[1], Peng Lu[1]*, Matti Leppäranta[2], Bin Cheng[3], Ruibo Lei[4], Bingrui Li[4], Qingkai Wang[1], Zhijun Li[1]*

[1]State Key Laboratory of Coastal and Offshore Engineering, Dalian University of Technology, Dalian, China

[2]Institute of Atmosphere and Earth Sciences, University of Helsinki, Helsinki, Finland

[3]Finnish Meteorological Institute, Helsinki, Finland

[4]MNR Key Laboratory for Polar Science, Polar Research Institute of China, Shanghai, China

*Corresponding to:* Peng Lu (lupeng@dlut.edu.cn), Zhijun Li (lizhijun@dlut.edu.cn).

**Abstract.** Variations in Arctic sea ice are not only apparent in its extent and thickness but also in its internal properties under global warming. The microstructure of summer Arctic sea ice changes due to varying external forces, ice age, and extended melting seasons, which affect its optical properties. Sea ice cores sampled in the Pacific sector of the Arctic obtained by the Chinese National Arctic Research Expeditions (CHINARE) during the summers of 2008 to 2016 were used to estimate the variations in the microstructures and inherent optical properties (IOPs) of ice and determine the radiation budget of sea ice based on a radiative transfer model. The variations in the volume fraction of gas bubbles ($V_a$) of the ice top layer were not significant and $V_a$ of the ice interior layer was significant. Compared with 2008, the mean $V_a$ of interior ice in 2016 decreased by 9.1%. Meanwhile, the volume fraction of brine pockets increased clearly during 2008-2016. The changing microstructure resulted in the scattering coefficient of the interior ice decreasing by 38.4% from 2008 to 2016, while no clear variations can be seen in the scattering coefficient of the ice top layer. These estimated ice IOPs fell within the range of other observations. Furthermore, we found that variations in interior ice were significantly related to the interannual changes in ice ages. At the Arctic basin scale, the changing IOPs of interior ice greatly changed the amount of solar radiation transmitted to the upper ocean even when a constant ice thickness is assumed, especially the thin ice in marginal zones, implying the presence of different sea ice bottom melt processes. These findings revealed the important role of the changing microstructure and IOPs of ice in affecting the radiation transfer of Arctic sea ice.

## 1    Introduction

The recent rise in air temperature in the Arctic is almost twice the global average, known as Arctic amplification (Dai et al., 2019), which has been seen in the retreat of sea ice, especially in summer. The extent of sea ice in summer has decreased (Comiso et al., 2008; Parkinson and Comiso, 2013; Petty et al., 2018), and summer ice is thinner (Kwok, 2018), younger (Stroeve and Notz, 2018), and warmer (Wang et al., 2020) than before. These changes have affected the transfer of sunlight

into the Arctic Ocean, and the optical properties of sea ice are changing the solar radiation budget in the area.
Variations of Arctic sea-ice cover are related not only to the macroscale properties described above but also to the ice
microstructure. Sea ice is a multiphase medium consisting of pure ice, gas bubbles, brine pockets, salt crystals, and sediments
(Hunke et al., 2011). In the last decades, the length of the Arctic ice melt season has shown a significant positive trend (Markus
et al., 2009), and the Arctic ice cover has experienced a transition from predominantly old ice to primarily first-year ice
(Tschudi et al., 2020; Stroeve and Notz, 2018). At the same time, in melting ice gas bubbles and brine pockets tend to become
larger (Light et al., 2003), and phase changes due to brine drainage and temperature result in variations in the volume of gas
and brine (Crabeck et al., 2019; Weeks and Ackley, 1986). Except for the above-mentioned factors, absorption of shortwave
radiation, synoptic weather, and surface melt pooling can also partly affect the ice microstructure. Therefore, the physical
properties of ice have changed and in the past 10 years the bulk density of summer Arctic sea ice has been lower than reported
in the 1990s due to increased ice porosity (Wang et al., 2020). Despite the changing ice microstructure having attracted
attention, there is still no quantitative description of its evolution and effect factors (Petrich and Eicken, 2010).
Gas bubbles and brine pockets, as dominant optical scatterers, directly influence the inherent optical properties (IOPs) of
sea ice (Grenfell, 1991; Perovich, 2003a). IOPs include scattering and absorption coefficients and information about the phase
function of the domain. The varying IOPs of ice have attracted attention due to their important role in the process of light
penetration in ice. Light et al. (2008) and Katlein et al. (2019; 2021) demonstrated clear different IOPs in sea ice of different
depths. The differences in the IOPs between first-year ice and multiyear ice have been ascertained in many observations (e.g.,
Light et al., 2015; Grenfell et al., 2006). There are also some differences in the bulk IOPs of first-year ice because of the
different stages of melting (Veyssière et al., 2022). However, the available observed or estimated ice IOPs were rare, which
resulted in quantitative knowledge of the progression of the sea ice IOPs and their influencing factors was still absent (light et
al. 2015). Even in the latest studies and sea ice models, IOPs are set as constants based on previous field observations (Briegleb
and Light, 2007), which is somewhat in contrast to the reality in the Arctic Ocean.
Changes in ice microstructure or IOPs are especially important for the energy budget of Arctic ice under the general
warming climate and decreasing ice age. The reason for this is their direct effect on ice apparent optical properties (AOPs),
which influence the partitioning of radiation in the Arctic by various feedback processes. However, the observed relationships
between ice microstructure, IOPs, and AOPs are rare in the available literature. Parameterization proposed by Grenfell (1991)
was the most widely used method to estimate the response of ice IOPs to microstructure. Due to the lack of detailed, observed
ice microstructure, this method was usually used to build models (Light et al., 2004; Yu et al., 2022; Hamre, 2004). In the
latest MOSAiC expedition during 2019-2020, Smith et al. (2022) observed the formation of a porous surface layer (i.e. surface

scattering layer, SSL) of sea ice and its enhancement on ice albedo. Macfarlane et al. (2023) further in detail described the microstructure of SSL using X-ray tomography and its effects on ice optical properties. They are the first to link ice microstructure and optical properties by field observations.

In this study, *in situ* observations of the physical properties of summer Arctic sea ice during the Chinese National Arctic Research Expeditions (CHINARE) from 2008 to 2016 were employed as input data. Variations of the microstructure and the IOPs of Arctic sea ice are presented. Also shown are their quantitative effects on the radiation budget. Applying these varying IOPs to satellite-observed sea ice conditions has allowed us to estimate the role of ice microstructure in the radiation budget in the Arctic basin scale.

## 2    Data and method

### 2.1    Arctic sea ice coring

The Arctic sea ice cores were sampled in the Pacific sector of the Arctic Ocean during summer cruises of the CHINARE program from 2008 to 2016 (Figure 1). The ice cores in each year were composed of different numbers of first-year ice and multiyear ice with thicknesses from 0.6 to 1.9 m. Detailed volume fractions of the gas bubbles and brine pockets ($V_a$, $V_b$) in the ice cores were given by Wang et al. (2020). The mean sampling date of ice cores was Aug. $20 \pm 8$ days, when the ice had been melting for a while (~59 days) and had not yet begun to freeze according to the melting onset data from NASA. According to previous observations, SSL of sea ice can be re-formed within a couple of days after removal (Smith et al., 2022). There are no clear temporal changes in the microstructure of surface ice in the entire July (Macfarlane et al., 2023). Furthermore, the ice surface melt rate in August was only ~1/10 of that in July (Nicolaus et al., 2021; Perovich, 2003b). That is, it is expected that the microstructure of the ice surface was similar in the mid- and late-melting seasons that cover the sampling dates of the present ice cores. Therefore, the short-term temporal variability of ice cores was expected not to affect their surface ice microstructure.

To further reduce the impact of temporal variations in the ice cores on the ice microstructure, we preprocessed the ice core data. The ice cores in each year were allocated different weights according to their sampling date. The weight ($w$) of ice cores in affecting period ($D$) can be obtained according to the Cressman method: $w = \frac{D^2 - d^2}{D^2 + d^2}$, where $d$ is the number of days from the mean sampling date. Then the weighted mean of ice properties was $\bar{x} = \frac{\sum_{i=1}^{n} w_i x_i}{\sum_{i=1}^{n} w_i}$. $D$ was set to 30 days because the weighted mean values and deviations were nearly unaffected when the $D$ was over 30 days. In the following analyses, the mean values of each year refer to the weighted ones. After the preprocessing, the deviation of melting days in a single year

was reduced by ~50.5%. As for the spatial variations in the ice cores, it is difficult for field observations to avoid the effects
of spatial variations. Therefore, related studies have generally ignored the effects of sampling locations on the statistics (Carnat
et al., 2013; Frantz et al., 2019; Katlein et al., 2019; Light et al., 2022). Related discussion about the temporal and spatial
variations can be found in Section 4.2.

A typical undeformed sea ice floe consists texturally of three layers due to its growth conditions (Tucker et al., 1992).

The first two layers are relatively thin and consist of a granular layer and a transition layer, and the lowest layer generally
consists of columnar ice. The ice texture controls the ice microstructure (Crabeck et al., 2016). Thus, the development of gas
bubbles, brine pockets, and IOPs in the three ice layers is different. Analogous to the parameterization of the Los Alamos sea
ice model (CICE; Briegleb & Light, 2007), Each ice core was evenly divided into 10 layers. The top (1/10) layer of an ice core
was defined as the top layer (TL), the second layer (2/10) was the drained layer (DL), and layers 4–10/10 collectively
constituted the internal layer (IL). Note that the surface scattering layer (SSL) and part of the DL were mixed in the TL and
could not be separated completely. Layer 3/10 was also a mixture of a DL and IL, and is therefore neglected in the following
analysis.

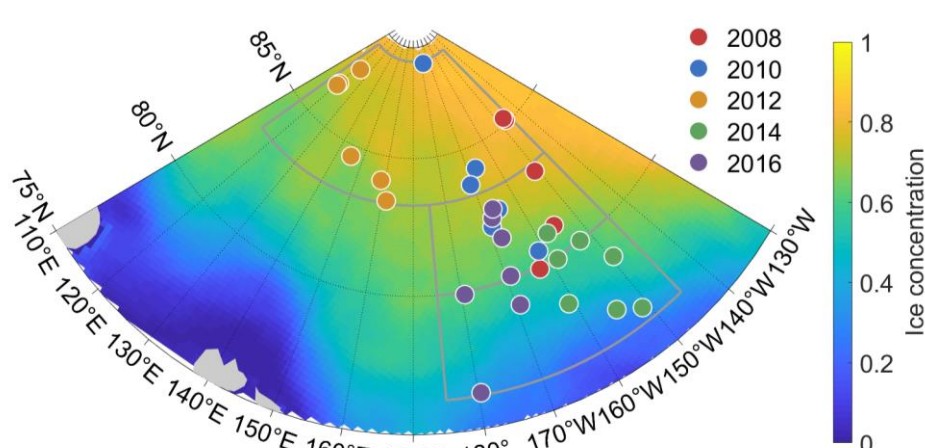


Figure 1. Locations of the sampled ice cores during CHINARE cruises. The ice cores were assorted into three parts according to latitude and
ice concentration. Their quantities were nearly the same in each zone. The ice concentration in the base map was the mean in August from
2008 to 2016.
**2.2   Sea ice optics modeling**

The IOPs of sea ice, including the scattering coefficient, $\sigma$, absorption coefficient, $\kappa$, and asymmetry parameter, $g$, can be

determined directly from the ice microstructure. Following the theory of Grenfell (1991), scattering in ice is caused by gas
bubbles and brine pockets, and absorption is caused by brine pockets and pure ice. This parameterization has been proved by
extensive observations (Light et al., 2004; Smedley et al., 2020). The IOPs of sea ice can be obtained from the sum of the
scatterers weighted by their relative volumes as:
$$\sigma = \sigma_a + \sigma_b = \int_{r_{min}}^{r_{max}} \pi r_a^2 Q_a^{sca} N_a(r) dr + \int_{l_{min}}^{l_{max}} \pi r_b^2 Q_b^{sca} N_b(l) dl \qquad (1)$$
$$\kappa = \kappa_i + \kappa_b = k_i V_i + \int_{l_{min}}^{l_{max}} \pi r_b^2 Q_b^{abs} N_b(l) dl \qquad (2)$$
$$g = \frac{g_a \sigma_a + g_b \sigma_b}{\sigma} \qquad (3)$$
In these equations, the subscripts a and b represent gas bubbles and brine pockets, respectively, $r$ is their radius (or equivalent
radius), and $l$ is the length of the brine pockets. $Q^{sca}$ and $Q^{abs}$ are the scattering and absorption efficiencies, respectively, which
can be calculated using Mie theory. $N$ is the size distribution function, subscript i represents pure ice, and $V_i = 1 - V_a - V_b$ is its
volume fraction. The values of these parameters are summarized in Table 1. Brine pockets longer than 0.03 mm are modeled
as cylinders rather than spheres (Light et al. 2003). The conversion function from Grenfell & Warren (1999) is employed to
represent hexagon columns as spheres with the same optical properties. Besides, $Q^{abs}$ and $Q^{sca}$ in the required size range are
obtained using their effective radii, which are calculated according to Hansen & Travis (1974).

Table 1. Parameters used in the radiation transfer model in Arctic summer and their sources

| Parameter | Reference(s) |
|---|---|
| refractive index of gas bubbles | Light et al. (2004) |
| refractive index of brine pocket | (Smith and Baker, 1981) |
| $N_a$, $N_b$ | Light et al. (2003) |
| $k_i$ | Grenfell and Perovich(1981) |
| $g_a$, $g_b$ | Light et al. (2004) |
| $r_{min} = 0.5$ mm, $r_{max} = 2$ mm | Grenfell(1983); Frantz et al. (2019) |
| $l_{min} = 1$ mm, $l_{max} = 20$ mm | Light et al. (2003); Frantz et al. (2019) |


The Delta-Eddington multiple scattering model, where the constant IOPs from Briegleb & Light (2007) were replaced by
the modeled IOPs, was employed to estimate the apparent optical properties (AOPs: albedo $\alpha_\lambda$, transmittance $T_\lambda$, and
absorptivity $A_\lambda$) of the ice at the sampling sites (Yu et al., 2022). This radiative transfer model was commonly used, and its
accuracies were widely accepted. The integrated albedo ($\alpha_B$), transmittance ($T_B$), and absorptivity ($A_B$) were calculated by
integrating the spectral values over the band of the incident solar radiation, $F_0$ as:
$$X_B = \frac{\int_{\lambda_1}^{\lambda_2} X_\lambda F_0(\lambda) d\lambda}{\int_{\lambda_1}^{\lambda_2} F_0(\lambda) d\lambda}, X = \alpha, T, A, \qquad (4)$$
In the following sections, the integrated absorption coefficient, $\kappa_B$, was also derived by this equation, following CICE
(Briegleb & Light, 2007). Considering the generally cloudy weather in Arctic summer, the incident solar irradiance under an
overcast sky in August from Grenfell & Perovich (2008) was chosen as the default value for $F_0$. The studied wavelength band
was set as the photosynthetically active band, i.e. $\lambda_1 = 400$ nm and $\lambda_2 = 700$ nm.

## 2.3 Arctic-wide up-scaling

To conduct an up-scaling analysis of the radiative budget of the Arctic sea ice cover based on observations of the ice
microstructure in the Pacific sector, we used representative basin-scale sea ice data to estimate the variations in the distribution
of radiation fluxes in summer during 2008-2016. The sea ice concentration ($C$) was provided by the National Snow and Ice
Data Center (NSIDC) (DiGirolamo et al., 2022), the sea ice thickness was based on CryoSat-2/SMOS data fusion (Ricker et
al., 2017), and the downward shortwave radiation flux at the surface ($E_d$) was obtained from the European Centre for Medium-
Range Weather Forecasts (ECMWF). The latter two datasets were interpolated to a 25 km NSIDC Polar Stereographic grid.
Then, the mean radiation fluxes and ice concentrations from July to September from 2008 to 2016 were set as the representative
values in summer. Due to the limitation of satellite remote-sensing data of summer ice thickness, the representative thickness
was estimated according to the mean value in October from 2011 to 2016, together with the growth rate estimated by Kwok
and Cunningham (2016). Then, representative ice thickness can be obtained. These grided ice thickness and IOPs profiles from
ice cores were inputted in the radiative transfer model to estimate the ice AOPs. From all these data sets and the derived
parameters, the reflected, absorbed, and transmitted radiation flux by Arctic sea ice were calculated as $E_r = E_d \cdot C \cdot \alpha_B$, $E_a =$
$E_d \cdot C \cdot A_B$, and $E_t = E_d \cdot C \cdot T_B$, respectively.

## 3 Results

### 3.1 Microstructure of the ice cores

There were different variation trends in the volume fraction of gas bubbles and brine pockets ($V_a$, $V_b$) as a function of ice
core depth (Figure 2). The upper granular ice was typically bubbly, associated with the drainage of brines, and the interior
columnar ice is usually depleted in gas bubbles (Cole et al., 2004). Thus, a significantly different $V_a$ could be seen (Analysis
of variance (ANOVA), $P < 0.01$) with a decreasing trend along depth (Pearson correlation coefficient, $r = -0.97$, $P < 0.01$).
The mean $V_a$ of the TL, DL, and IL for all ice cores was $23.4 \pm 5.6\%$, $17.9 \pm 5.3\%$, and $11.6 \pm 5.9\%$, respectively. These values
are similar to the observations made by Eicken et al. (1995) where $V_a$ decreased from $> 20$ % at the top to $< 5$ % at the bottom
for summer Arctic sea ice.
The different $V_b$ between layers was significant (ANOVA, $P < 0.01$). The drainage of brine resulted in a relatively small
$V_b$ of TL, with a mean of $3.5 \pm 2.4\%$, while it was $4.6 \pm 3.1\%$ and $13.5 \pm 6.7\%$ in the other two layers, respectively (Figure
2a). $V_b = 5\%$ is usually chosen as a threshold where discrete brine inclusions start to connect and the columnar ice is permeable
enough to enable drainage (Carnat et al., 2013). Thus, the ice cores in the present study have been melting for some time,
agreeing with the sampling season during CHINARE. Most $V_b$ profiles had a maximum in the middle depth, except for the ice
cores in 2012 (Figure 2d). This can be explained by the later sampling date in 2012 relative to the other years by about 10 days,
which resulted in enhanced brine drainage. Furthermore, the shape of the $V_b$ profile was also associated with the ice age (Notz
and Worster, 2009). Compared with the ice cores in 2010, although the ice cores in 2016 had similar sampling dates (one day
difference), the maximum position of $V_b$ in 2016 was lower than in 2010 (Figure 2c, f). This was because all ice cores in 2010
were sampled from first-year ice, and the ice cores in 2016 were comprised of first-year ice and multiyear ice (Wang et al.,

2020).

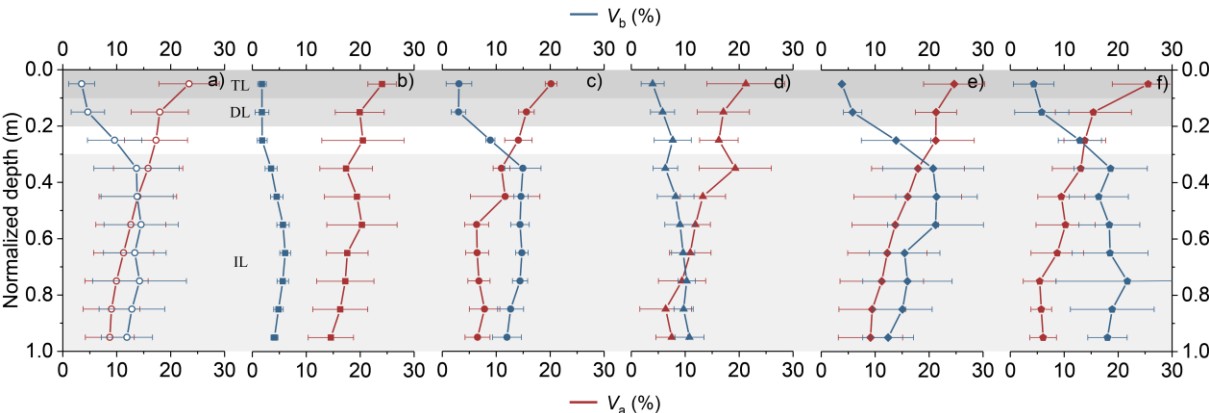


Figure 2. Profiles of $V_a$ and $V_b$ against normalized depth in (a) the whole study period, (b) 2008, (c) 2010, (d) 2012, (e) 2014, and (f) 2016.
The error bars show the standard deviation from the mean of the results. The shady areas represent the ice layer structure.

In addition to the different variations in $V_a$ and $V_b$ with depth, the annual variations in each layer were also different
(Figure 3a). $V_a$ was relatively small in the TL of 2010 because all ice cores were sampled from first-year ice (Wang et al.,
2020). The quantities of first-year ice cores were similar to the amount of multiyear ice cores in the other years. The variation
in $V_a$ of TL between years was statistically insignificant (ANOVA, $P > 0.1$). This indicated that the melting process of the ice
surfaces of the cores in different years was not different significantly. Contrary to the TL, the $V_a$ in the IL was different
significantly (ANOVA, $P < 0.05$). Compared with 2008, the mean $V_a$ of IL in 2016 decreased by 9.1%. The $V_a$ values of DL
were relatively stable and did not show significant variations in the study period.
Things were different for $V_b$ and ice porosity. There were increases in the mean $V_b$ of all three ice layers (Figure 3b).
Furthermore, the increases of mean $V_b$ in the IL were statistically significant ($r = 0.84$, $P < 0.1$; ANOVA, $P < 0.01$). From 2008
to 2016, the increase in the mean $V_b$ of IL was 13%. Simultaneously, the ice salinity of the IL decreased (Figure S1), which
agreed well with the observed and modeled results with warming conditions (Vancoppenolle et al., 2009). From the combined
effects of changing $V_a$ and $V_b$, there are no significant differences in the porosity of three layers (ANOVA, $P > 0.1$). Furthermore,

 the developments of porosity in the three layers are also similar (Figure 3c). Among the three layers, the statistical significance

of changing porosity of IL between years was relatively good (ANOVA, $P < 0.1$).

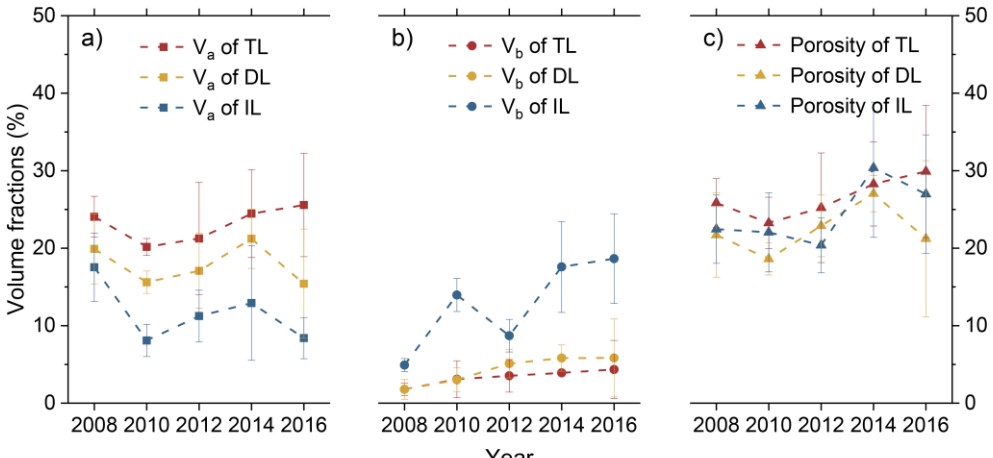


Figure 3. Variations in (a) $V_a$, (b) $V_b$, and (c) the porosity of the TL, DL, and IL of the ice cores during 2008-2016. The error bars show the
standard deviation for each year.

## 3.2 Variations in the IOPs of the ice cores

The mean scattering coefficient, $\sigma$, of the TL, DL, and IL for all ice cores was $264.5 \pm 26.7$ m$^{-1}$, $208.9 \pm 26.5$ m$^{-1}$, and
$160.9 \pm 33.3$ m$^{-1}$, respectively (Figure 4a). There was a significant decreasing tendency along with depth in the mean $\sigma$ of all
ice cores ($r = -0.97$, $P < 0.01$; ANOVA, $P < 0.01$), associated with a decreasing volume of gas bubbles (Figure 2). Although
the $V_b$ values of the ice cores increased clearly with depth, their effects on ice $\sigma$ were covered by the decreasing $V_a$. The reason
for this was that the refractive indices of brine pockets and pure ice are close (Smith and Baker, 1981; Grenfell and Perovich,
1981), which results in the effects of brine pockets on ice $\sigma$ were relatively weak than the gas bubble.
The vertical variations in $\kappa_B$ and $g$ were not clear as seen for $\sigma$ because they depend on $V_i$ and $V_b/V_a$, respectively. Due to
the effects of the ice porosity ($V_a + V_b$), $\kappa_B$ didn't show a statistically significant trend with depth (ANOVA, $P > 0.1$), which
varied in the range 0.09–0.1 m$^{-1}$. The mean value of $g$ was 0.93 except in 2008 (which was $g = 0.89$), and it significantly
increased with depth ($r = 0.91$, $P < 0.01$; ANOVA, $P < 0.01$). This value is similar to the commonly used one; for example,
the previous typical range of $g$ was from 0.86 to 0.99 (Ehn et al., 2008), and 0.94 was often adopted for computational efficiency
in models (Light et al., 2008). We note that the volume of brine pockets in ice cores of 2008 is relatively small, which was a
reason for the different values of $g$ found here.

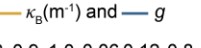

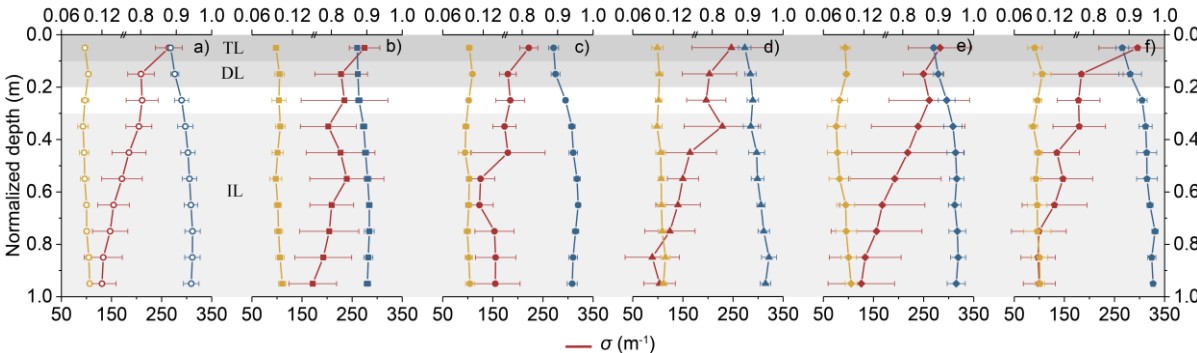

Figure 4. IOP profiles of ice cores against normalized depth in (a) the whole study period, (b) 2008, (c) 2010, (d) 2012, (e) 2014, and (f) 2016. The error bars show the standard deviation from the mean of the results.

The annual mean IOPs of the TL, DL, and IL of the ice cores are shown in Figure 5. As shown in Figure 5a, the variations in $\sigma$ of the TL, DL, and IL were different. The variation in $\sigma$ of the TL between years was statistically insignificant (ANOVA, $P > 0.1$), which reveals the relatively stable scattering ability of the ice surface. Things were different for IL, there were statistically significant variations in their $\sigma$ between years (ANOVA, $P < 0.05$). Compared with 2008, the $\sigma$ of the IL in 2016 decreased by 38.4% due to the decreased $V_a$ (Figure 3). The overall variations in the $\sigma$ of the DL were similar to that seen in the IL. Whereas, the former variations were not as clear as the latter due to ongoing drainage, and were not significant (ANOVA, $P > 0.1$).

There were no statistically significant differences in the integrated absorption coefficient, $\kappa_B$, of the TL, DL, and IL (ANOVA, $P > 0.1$), indicating the absorptivity of ice in different depths is similar. Furthermore, the developments of $\kappa_B$ in the three layers are similar (~ 0.001/year, Figure 5b). Among the three layers, the statistical significance of changing $\kappa_B$ of IL between years was relatively better (ANOVA, $P < 0.05$) than TL and DL. As shown in Figure 5c, the values of $g$ of the TL and DL were nearly constant. Because their values of $V_b$ were sufficiently small and similar due to drainage (Figure 3b), their values of $g$ are mainly attributed to gas bubbles. In contrast, the $g$ of IL varied significantly (ANOVA, $P < 0.01$). The values of $g$ of the IL increased by 5% with increasing $V_b$ in the study years (Figure 3b).

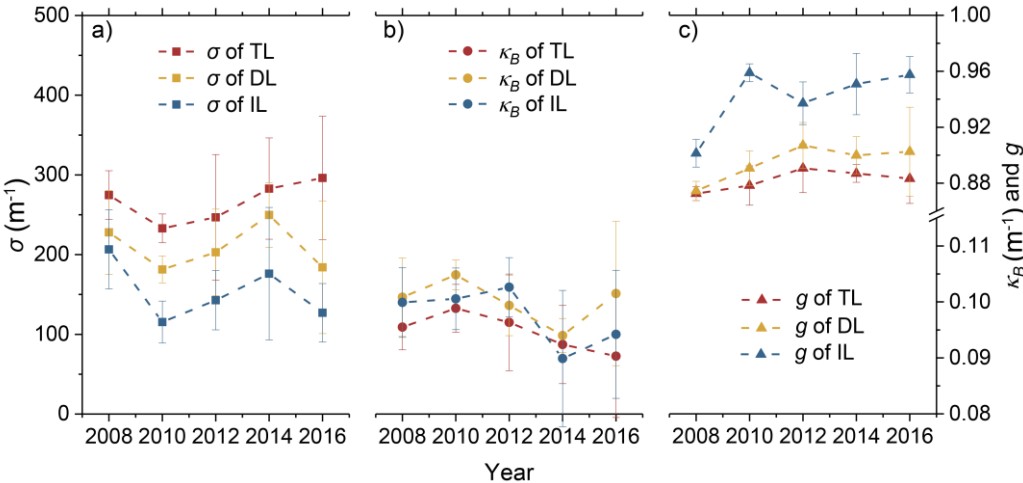


Figure 5. Annual (a) $\sigma$, (b) $\kappa_B$, and (c) $g$ for the TL, DL, and IL of the ice cores from 2008 to 2016. The error bars show the standard deviation
in each year.

### 3.3 Variations in the AOPs of the ice cores

Having seen that the IOP profiles of the sea ice were not constant in the different years (Figure 5), a more important
question is how these changes affected the AOPs. The radiative transfer model was employed here to estimate the AOPs of
sampling sites, as shown in Figure 6. Note that the AOPs here were calculated based on the level ice. Surface properties, such
as a snow layer or melt ponds, were not considered here, because the focus was on the effects of the ice microstructure on their
AOPs. The results obtained with the same IOPs profiles but for a constant reference ice thickness (1 m) are also presented to
quantify the contributions from the ice microstructure and thickness separately. This reference thickness was chosen to study
the vertical structure relation to the surface and bottom and compare the samples with different thicknesses. This doesn't affect
the trends in Fig 6.
It can be seen from Figure 6a that the thickness of ice cores decreased in study years with a statistically significant trend
($r$ = -0.89, $P$ < 0.05) and variations (ANOVA, $P$ < 0.05). The values of $\alpha_B$ changed because of the effects of the ice IOPs and
thickness (Figure 6b). The variations in mean $\alpha_B$ during 2008-2014 were similar to those in the $\sigma$ of the TL and DL. In 2016,
the mean $\alpha_B$ decreased due to the decreasing ice thickness. As a result, there are no statistically significant variations in $\alpha_B$
between years (ANOVA, $P$ > 0.1). This was different from the remote-sensing results (-0.05 per decade from 1982 to 2009) of
Lei et al. (2016). Part of the reason for this was the direct factor that reduces the annual ice albedo is not the ice microstructure
but rather the surface conditions. Eicken et al. (2004) and Landy et al. (2015) reported that the evolution of melt ponds on the
ice surface could explain 85% of the variance in the summer ice albedo.
Different from $\alpha_B$, annual variations in $T_B$ and $A_B$ were significant (ANOVA, $P$ < 0.05). The $T_B$ ($A_B$) tended to increase

(decrease) with years (Figure 6c). The mean value of $T_B$ in 2016 was over treble of that in 2008. Meanwhile, $A_B$ decreased by

about 19.5% from 2008 to 2016. Furthermore, the change of $A_B$ in the study years was lower than the actual change in the ice

thickness (-35.0%). Thus, the difference, 23.8% ($\frac{1-19.5\%}{1-35.0\%} - 1$), was attributed to an increase in the absorbed solar energy per

unit volume of sea ice. This result does match the findings of Light et al. (2015), which showed that the thickness of first-year

ice was less by 13.3% than multiyear ice (1.3 m vs. 1.5 m, respectively). However, the radiation absorbed by the former was

less by 2% than the latter. In other words, the solar energy absorbed by a unit volume of first-year ice was greater than multiyear

ice by 12.5%.

To make a direct comparison with the above variations, we considered a constant ice thickness, finding no clear changes

in $\alpha_B$ (Figure 6b). Meanwhile, the variations in $T_B$ and $A_B$ were different clearly with similar overall trends (dashed lines in

Figure 6c). $T_B$ increased from 0.03 to 0.07 from 2008 to 2016, accounting for about 33.1% of the real change ratio with

changing thickness. Thus, the changing microstructure of the melting ice resulted in an increased transmittance that was

independent of the ice thickness. A similar result was observed in the laboratory, where the changing ice microstructure during

the warming process (no decrease in thickness) increased the ice transmittance (Light et al., 2004). Different from $T_B$ and $A_B$,

whether the thickness was accounted for or not, the variations in $\alpha_B$ were hardly affected. This demonstrated that the present

variations in ice thickness had more effects on the ice $T_B$ and $A_B$ than $\alpha_B$.

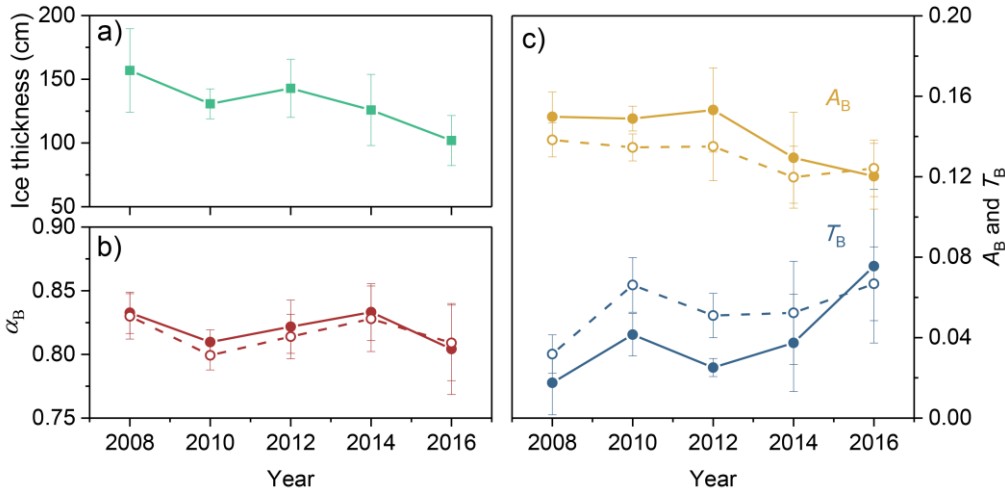

Figure 6. (a) Thickness and (b, c) estimated AOPs of the ice cores from 2008 to 2016. Also shown as dashed lines are the AOPs with the
same IOPs and constant thickness (1 m). The error bars show the standard deviation in each year.

### 3.4 Arctic-wide estimation

It may be interesting to estimate the quantitative effects of varying IOPs on the radiation distribution of the Arctic with a

real ice thickness field, we expand the variations of the ice cores (Figure 5) to an Arctic-wide scale under the following

assumptions. 1), the IOPs of Arctic ice can be represented by our ice cores data. They are taken as constant, and seasonal and
spatial differences are ignored. This is justified since such a hypothesis has been widely used (Briegleb and Light, 2007). 2), a
decreasing trend of -5.8 cm yr$^{-1}$ in ice thickness according to Lindsay and Schweiger (2015) was adopted to get a general view
of the contributions of the changing ice thickness on the radiation budget. The representative basin-scale sea ice and radiation
data in summer (see Section 2.3) were used here to estimate the variations in the distribution of radiation fluxes.
With the combined effects of the changing microstructure and thickness of ice, Arctic-wide variations in the mean $\alpha_B$, $T_B$,
and $A_B$ were statistically significant (ANOVA, $P < 0.01$) and clearer than those in Figure 6 (Figure 7a), especially the overall
trends of the mean $T_B$ ($r = 0.95$, $P < 0.01$) and $A_B$ ($r = -0.98$, $P < 0.01$) of ice. Although the mean $\alpha_B$ decreased from 2008 to
2016, there was not much change in reflected solar flux ($E_r$), about 51.2 W m$^{-2}$ during the study years (Figure 7b). This was
resulted from that the decreasing $\alpha_B$ was largely provided by marginal ice zones. The decreasing rate of $\alpha_B$ in regions with ice
thicknesses < 1 m (equivalent to 16.4% of the entire ice area) was over 1.6 times the rate of the entire ice cover (Figure S2).
With the retreat of sea ice, the reflected flux of the marginal zone contributes less and less to the reflected flux of the entire ice
cover.
Different from $E_r$, the overall trends of transmitted ($E_t$) and absorbed solar flux ($E_a$) were clear under the combined effects
of the changing microstructure and ice thickness. The mean $E_t$, was significantly different between years (ANOVA, $P < 0.01$),
and increased from 1.8 W m$^{-2}$ to 9.0 W m$^{-2}$ from 2008 to 2016 significantly ($r = 0.93$, $P < 0.05$, Figure 7b). Most of the increase
in $E_t$ is ascribed to thin ice in marginal ice zones (ice thicknesses < 1 m), which contributed 51.8% of the increasing $E_t$ from
2008 to 2016 (Figure 8a–e). Meanwhile, variations in transmitted solar radiation $E_a$ were significant (ANOVA, $P < 0.01$). The
$E_a$ decreased from 8.6 W m$^{-2}$ in 2008 to 7.2 W m$^{-2}$ in 2016 significantly ($r = -0.94$, $P < 0.05$). As the decrease in ice volume
from 2008 to 2016 was 32.2%, the solar energy absorbed by a unit volume of sea ice increased by 23.4% on the Arctic scale.

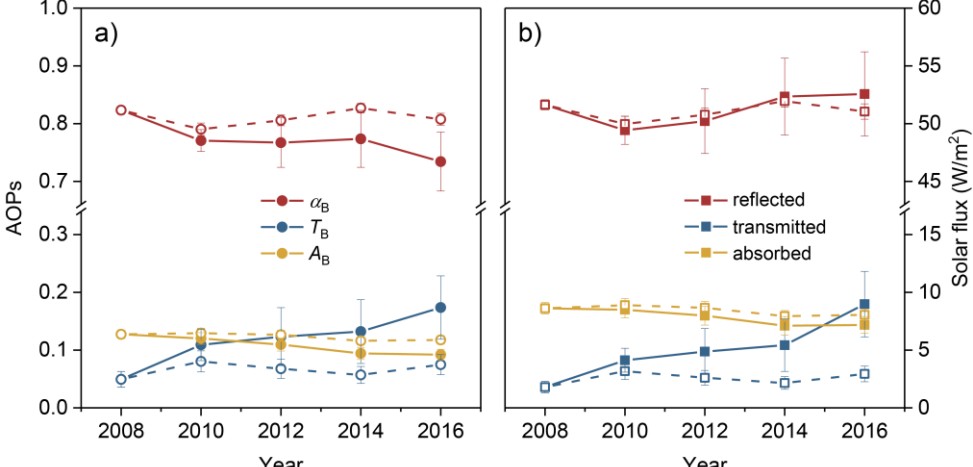

Figure 7. Arctic-wide variations in the mean (a) AOPs of ice and (b) solar flux distribution during 2008-2016. Also shown as dashed lines are the AOPs and fluxes with the same IOPs and constant thickness field. The error bars show the standard deviation in each year.

When the ice thickness was set as a constant, variations in the mean AOPs were different, which resulted in differences in the solar flux (dashed lines in Figure 7b). Among them, differences in the reflected flux $E_r$ were relatively small. Meanwhile, the mean $E_t$ increased from 1.8 W m$^{-2}$ in 2008 to 2.9 W m$^{-2}$ in 2016, with no significant trend. $E_a$ decreased from 8.6 W m$^{-2}$ to 8.0 W m$^{-2}$ in the same period. These changes corresponded to 16.0% and 39.3% of the combined effects of the ice IOPs and thickness, respectively, from 2008 to 2016. Furthermore, marginal ice zones with ice thicknesses < 1 m still contributed 38.5% of the increasing $E_t$ from 2008 to 2016 (Figure 8f-j). This value was about 74.3% of the rate of the combined effects of the changing IOPs and thickness of ice. In other words, the same changes in the ice microstructure had more effects on the $T_B$ of thin sea ice, and these effects were clearer than those resulting from general decreasing ice thickness.

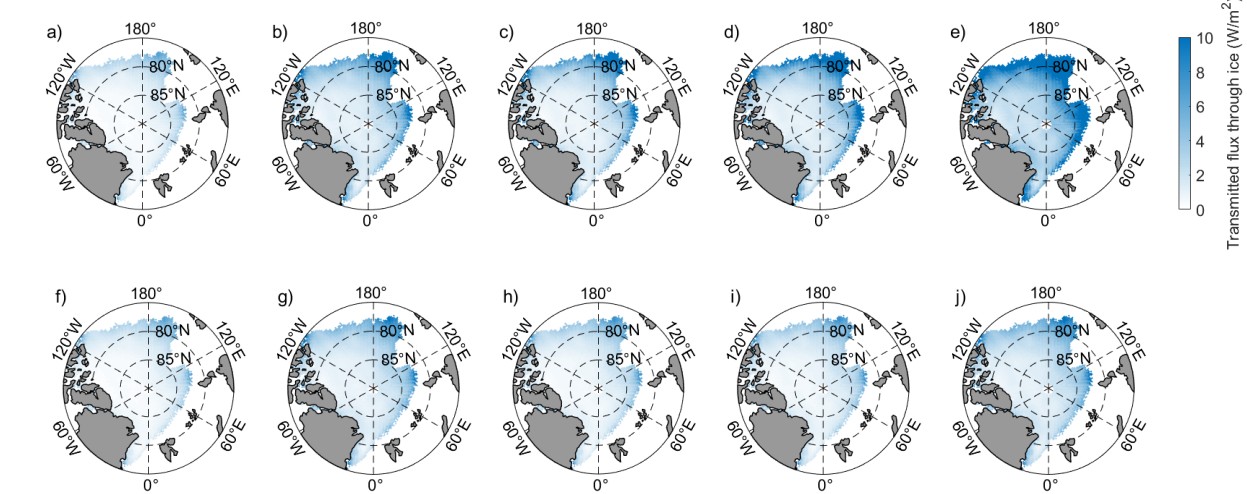

Figure 8. Distribution of transmitted solar radiation through sea ice in the summers of 2008 to 2016 when the sea ice thickness was set (a–e) to decrease and (f–j) to a constant value. Only flux that penetrated through the sea ice is considered in this map.

# 4    Discussion

## 4.1    Comparisons with IOP measurements

In Section 3.2, we estimated the ice IOPs according to the observed ice physics and structural-optical theory. Other methods were used to estimate ice IOPs in previous studies. In this section, we compare the ice scattering coefficient, the most variable value among IOPs, determined in the present study with previous results (Figure 9). It is difficult for us to consider the potential affecting factors because the variations in $\sigma$ were still unclear. So, we pay more attention to the comparison of $\sigma$ range. The differences in wavelength bands were ignored in the comparisons because $\sigma$ was nearly wavelength-independent.

It is clear from Figure 9 that the range of $\sigma$ of the present study covered the majority of previous results. The derived values of $\sigma$ for the SSL and DL of melting bare ice in August ranged from 920 to 2,000 m$^{-1}$ and 40 to 150 m$^{-1}$, respectively (Light et al., 2008). According to the layer structure, wherein the TL was composed of a 5 cm SSL and the others were DLs, the bulk $\sigma$ of the TL in Light et al. (2008) ranged from 270 to 435 m$^{-1}$. This result was slightly higher than our results. The results of Mobley et al. (1998) and Perron et al. (2021) agree with our range. The $\sigma$ of the DL in Perron et al. (2021) was in our range, and the values of Light et al. (2008) were smaller than those in the present study.

Differences in the $\sigma$ of the IL were clearer than in the TL and DL. The $\sigma$ values of the IL of most our cores were relatively larger than those of Light et al. (2008, 2015) and Frantz et al (2019). In these results, Light et al. (2008) estimated the $\sigma$ using the observed ice albedo and a three-layer structure with fixed thicknesses. The results of Light et al. (2015) and Frantz et al. (2019) were obtained in a cold laboratory by simulating the radiative transport in subsections of sea ice. Meanwhile, the results of Grenfell et al. (2006) and Perron et al. (2021) are close to the minimum of our range. The $\sigma$ of ice in Grenfell et al., (2006) was calculated from the ice extinction coefficient, and it was measured *in situ* using a diffuse reflectance probe in the Perron et al. (2021). The values calculated by the same method as used in the present study by Mobley et al. (1998) were close to the maximum of our range. Thus, it was expected that the differences in the IL's $\sigma$ partly resulted from the different methods used in the myriad studies.

One possible reason for the differences was the uncertainties in the ice microstructure introduced by brine loss during measurement and segmenting. Thus, our $V_a$ values of the IL are greater than the values derived from nondestructive methods (e.g., Perron et al., 2021). As a result, the maximum underestimate of $V_b$ was 15–25% and the maximum overestimate of $V_a$ was 96–160% when taking the uncertainties introduced by the measurements and brine drainage into account (Wang et al., 2020). Taking the mean $V_a$ and $V_b$ of all ice cores as an example, these uncertainties overestimated the $\sigma$ of the IL by 78 m$^{-1}$ at most. Although brine loss during sampling and measurements introduced uncertainties to $V_a$ and $V_b$, the methods used for obtaining and measuring the ice cores during the CHINARE cruises were the same. Therefore, the uncertainties introduced by

the methodology hardly affected the changes seen in Figure 6 and Figure 7.

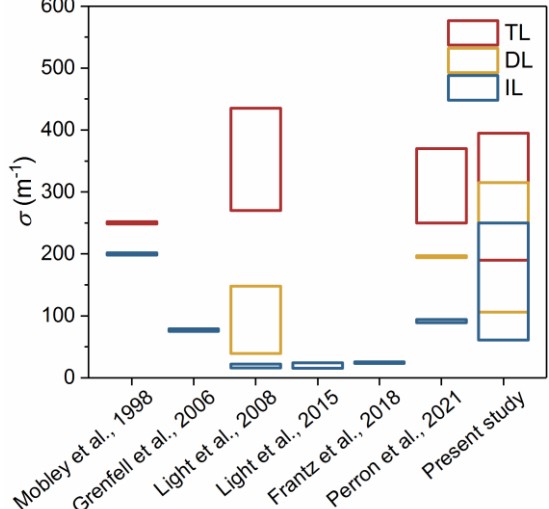


Figure 9. Comparison of the ice scattering coefficient in the present study to the published results for Arctic sea ice using various methods.
All comparison results have been scaled to the layer structure used in the current study according to their ice thicknesses.

Another source of difference is the distribution function of gas bubbles employed in the IOP parameterization. Many

distributions are obtained in a cold laboratory, where the ice temperature is not consistent with that in the summer Arctic. As
the refractive indices of brines and pure ice were similar, the distribution function of brine pockets had a smaller influence on
the ice IOPs than gas bubbles (Yu et al., 2022). Here, we tentatively adjusted the exponent of the distribution function of the
gas bubbles from its default value of -1.5 to -1, i.e., the fraction of small bubbles decreases, which coincides with warming ice
(Light et al., 2003). Then, the changed distribution function was used for 1 m thick ice with mean values of $V_a$ and $V_b$ for every
ice core. This change resulted in an uncertainty of 8 m$^{-1}$ in the $\sigma$ of each layer. These uncertainties did not alter the above
results and are considered acceptable.

Although brine loss and the difference in the distribution functions of gas bubbles introduced uncertainties in $\sigma$, they did

not affect the ice AOPs much. Considering a 1 m thick ice layer described by the mean physics of ice cores, the effects of the
former factor on the ice AOPs were less than 0.02. The uncertainties in $\alpha_B$ and $T_B$ introduced by the latter factor were 0.005
and 0.002, respectively. Therefore, our estimated $\alpha_B$ range (0.76–0.87) agreed with the observed results of Light et al. (2008,
2015) and Grenfell et al. (2006). Meanwhile, the estimated $T_B$ (0.01–0.1) was also in the corresponding observed ranges.

## 4.2  On the potential interannual variations of the IOPs

Extensive measurements of the IOPs of Arctic sea ice have been carried out, and some authors have noticed the seasonal variations of the ice microstructure and IOPs (Light et al., 2008; Frantz et al., 2019; Katlein et al., 2021). However, if there are interannual variations in sea ice IOPs are still not clear, although such changes in sea ice extent, thickness, and age are evident. A lack of continuous IOP measurements is the primary reason. Compared with previous observations, the ice core data in the present study were more appropriate for analyses on the potential interannual variations in ice IOPs because of their long time span and consistencies in the sampling method, seasons, and sea areas. The reason we could not introduce other ice core data (SHEBA, ICESCAPE, N-ICE, MOSAiC, etc.) into this study was that not only the differences in sampling seasons, sites, and methods increase the dispersion in time and space during such an analysis, but also the lack of information about the ice microstructure or essential physical properties will limit how much we can determine from such a comparison. We consider the presented ice core data is the best possible estimate on the potential interannual variations at this time, while acknowledging that further improvements of the data products are needed. Considering that sampling ice cores is a commonly used method for *in situ* observations, with more suitable ice core data in the future, large-scale time series of ice IOPs may be obtained.

The ice cores used in the present study were sampled at different ice stations but not at the same floe (Figure 1). That is, the data did not form a continuous observation in the strictest meaning. Thus, the variations shown in Section 3 can be regarded as the combined effects from three parts, i.e. spatial, temporal, and interannual variations. To do the discussion of interannual variability, it is necessary to first establish the spatial and temporal variability of ice cores. Figure 10 illustrates the different IOPs of the ice cores in three latitude zones, which shows that there are spatial differences in the present ice core data. Among the three IOPs, variations in $\sigma$ are the clearest (up to 20%, Figure 10a). The differences in $\kappa_B$ and $g$ in the different latitude zones were not more than 5% and 3%, respectively (Figure 10b, c). As a transition layer between the TL and IL, variations in the IOPs of the DL were more discrete than in the other two layers. For now, we have little quantitative knowledge of the progressions of the sea ice IOPs and their influencing factors in the available literature. In the following discussion, the $\sigma$ was set as the main content.

It can be seen from Figure 10a that there were no clear changes in the mean $\sigma$ of TL in different latitude zones. Therefore, we ignore the spatial variations in $\sigma$ of TL. We further discuss its whole variations in different years. The variability of the ice surface is directly related to the number of melt days. The melt days are affected by the radiation balance, water vapor, air temperature, and other factors (Persson, 2012; Crawford et al., 2018; Mortin et al., 2016). Figure 11a shows the data obtained from ECMWF, the downward longwave radiation was $300.2 \pm 4.0$ W/m$^2$ at the surface during the study years with no statistically significant trend ($r$ = -0.57, $P$ > 0.1). The total column vertically integrated water vapor was also similar (11.9 $\pm$

0.4 kg/m$^2$) with no significant trend ($r = -0.58$, $P > 0.1$). Different from the surface radiation, we found the observed air
temperature increased at a speed of 0.14 ℃/year ($r = 0.84$, $P < 0.1$, Figure 11a). This clear difference in the temperatures was
not an exception but a general circumstance in the Arctic during 2008–2016 (Collow et al., 2020). This could also be seen in
the reanalysis data of ECMWF, where the mean air temperature in the summer of the study area has been increasing gradually
(0.12 ℃/year, $r = 0.84$, $P < 0.1$). With the effects of several factors, the melting days of sampling sites, which were calculated
according to the sampling date and melt onset from Markus et al. (2009) were $59 \pm 7$ days (Figure 11a). Their variation between
years was statistically insignificant (ANOVA, $P > 0.1$). In other words, there are no significant differences in the surface melt
of the ice cores in different years.
Previous observations demonstrated that ice surface melt was relatively weak in August (Nicolaus et al., 2021; Perovich,
2003b). Macfarlane et al. (2023) further found that the SSL microstructure of melting ice has no temporal changes. Meanwhile,
the differences in longwave radiation and vapor between sampling sites in single years were relatively small (Figure 11a). So,
it is expected that the scattering coefficient of TL also has no clear seasonal variations. Whereas, an increasing scattering in
the SSL during melt season was found in Light et al. (2008). This seems contrary to the findings of Macfarlane et al. (2023),
but it is not. As stated in Light et al. (2008), the observed increase in scattering represents not only an increased scattering in
a fixed depth layer but also an increased physical depth of the SSL or increased scattering of the next ice layer, because the
modeled layer thickness was fixed. What was the same in the two studies was approximately constant albedo (or reflectance).
This agrees with the similar albedo in Figure 6b of the present study, i.e. small seasonal differences don't affect the reflectivity
of bare ice. For now, there was no theoretical explanation or quantitative description of the evolution of the microstructure of
the ice surface during the melt (Petrich and Eicken, 2010). It can be seen from the present result, the increasing air temperature
seems not the predominant affecting factor in the late melting season. In short, it is expected that the effects of temporal
variations on the microstructure and IOPs of the ice surface were relatively small. Considering the whole variations in
microstructure (Figure 3) and IOPs (Figure 5) were not significant, there are no clear temporal, spatial, or interannual variations
in the ice surface of the present ice core data.

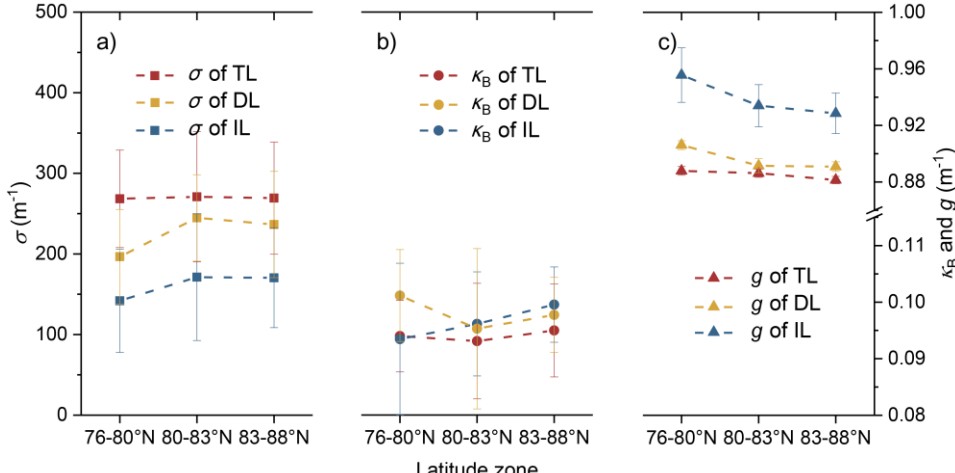

Figure 10. Different values of (a) $\sigma$, (b) $\kappa_B$, and (c) $g$ for the TL, DL, and IL of the ice cores in the three latitude zones. The error bars show the standard deviation in each latitude zone.

The $\sigma$ of the IL is relatively constant during the entire melt season (Light et al., 2008). That's to say, the whole variations in the ice interior layer didn't result from temporal factors. Meanwhile, the latitudinal differences in the $\sigma$ of the IL are clear. The $\sigma$ of the ice IL in the low-latitude zone was relatively smaller than that in mid- or high-latitude zones (Figure 10a). This is expected that the ice at lower latitudes is generally warmer earlier, which increases the brine inclusion size and connectivity of ice. Then naturally reduced the ice scattering coefficient. The spatial variation of mean $\sigma$ in the IL can be up to 30 m$^{-1}$ between low-latitude and mid- or high-latitude zones. This value was equivalent to 32.9% of the maximum of the whole variation. This implied that the spatial and interannual variations in ice properties together result in the changing IOPs shown in Figure 5. So, it is necessary to exclude the spatial variations before discussing the interannual changes of $\sigma$. According to the propagation law of variation, the square of whole variations of IL-$\sigma$ can be expressed as the square sum of their spatial variations and interannual variations. For the convenience of calculation, we ignored the small difference IL-$\sigma$ between mid- and high-latitude zones. There are five and three cores in 2014 and 2016 sampled in the low-latitude zone, respectively. According to the differences between ice cores from different years (whole variations, Figure 3) and different latitude zones (spatial variations, Figure 10a), we correct the mean $\sigma$ of the IL in 2014 from 176 m$^{-1}$ to 182 m$^{-1}$. That's to say, the interannual variations were larger than the whole variations by 6 m$^{-1}$. The value of 2016 was also corrected from 127 m$^{-1}$ to 131 m$^{-1}$ accordingly. Then, variations among the corrected $\sigma$ of the IL could be regarded as the result of the interannual factors.

Then, the corrected $\sigma$ of the IL was used to discuss the interannual changes. Figure 11b shows the correlations among the corrected $\sigma$ of the IL, ice age, and $T_B$ in study years. Also shown in circles were the uncorrected $\sigma$ of IL in 2014 and 2016. Note that $T_B$ here is the result under the assumption of a constant ice thickness (dashed line in Figure 6c). The ice ages were obtained

according to fieldwork (Wang et al., 2020) and remote-sensing data (Tschudi et al., 2019). Because the ice age of each grid
cell in the remote-sensing data is represented as the age of the oldest floe, once an ice core was distinguished as first-year ice
in the fieldwork, the corresponding ice age was set as one year regardless of the remote-sensing data. The use of remote-
sensing data is acceptable because the ice cores in this study were all sampled in large and thick floes for safe fieldwork. These
floes were more likely older than the surrounding ice. Figure 11b demonstrates that the decrease in the $\sigma$ of the IL is
significantly correlated with changing ice age ($r = 0.95$, $P < 0.01$). In other words, the ice age largely manifested in the ice
microstructure in the IL. A similar result was also observed i.e. the $\sigma$ of the IL in the first-year ice was smaller than in multiyear
ice (e.g. Light et al. 2015). This could also partly explain the spatial variations in the $\sigma$ of the IL (Figure 10a) because sea ice
in high-latitude zones was likely older than in the other zones (Stroeve and Notz, 2018). Furthermore, there are significant
correlations between $\sigma$ of the IL and ice $T_B$ ($r = -0.93$, $P < 0.05$). That's to say, the changing ice age can be responsible for the
modeled results of changing ice transmittance shown in Figure 7, even without any decrease in the ice thickness. One other
thing to point out, the changing ice age seems to not affect the albedo of bare ice (Figure 6b). Light et al. (2022) suggest that
the principal reason for this is the SSL shows invariance across location, decade, and ice age, which was confirmed by
comparing data from MOSAiC (2019-2020) and SHEBA (1997-1998). Our results partly prove this view i.e. there are
significant variations in the ice age but no significant variations in microstructure or IOPs of TL during 2008-2016.

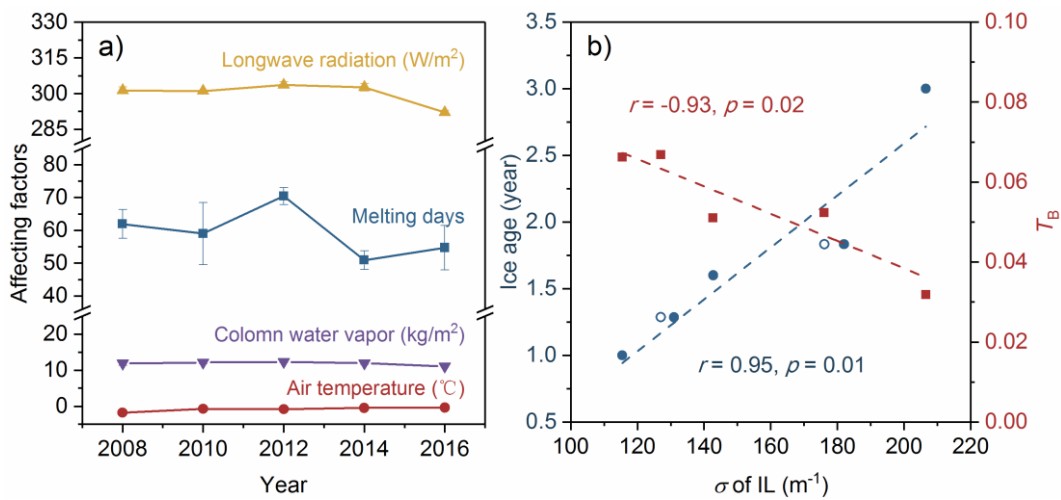

Figure 11. (a) Changing melting days, surface downward longwave radiation flux, total column vertically-integrated water vapor, and
observed air temperature at the sampling sites. The error bars show the standard deviation in each year. Some error bars are invisible
because they are small enough. (b) Correlations among the $\sigma$ of the IL with ice age and $T_B$. The circles denote the uncorrected data.

In summary, we didn't find significant variations in the IOPs of the ice top layer. Meanwhile, the differences in the IOPs
of the ice IL were related to interannual variations in the ice age. To our knowledge, this is the first study to link ice
microstructure and optical properties at interannual scales. Although these ice core data are not a time series in the strictest
meaning, they are still helpful for understanding the general effects of the scenario where the Arctic ice ages are decreasing.
Our results suggested that in this scenario, the $\sigma$ values of the IL of summer ice tended to be smaller than before. It is expected
to lead to interannual trends of the ice microstructure and IOPs. Then, more solar radiation transmits into the ocean. The effects
of this process need more attention in future observations and simulations.

### 4.3   Implications for the future Arctic

Previous studies have reported that surface properties (snow, ponds, etc.) largely control the variations in the ice albedo
(Landy et al., 2015). The present results also asserted that variations in the ice's microstructure or IOPs had little effect on the
albedo of bare ice ($< 2\%$), but they do play an important role in ice transmittance (Figure 6). With continued Arctic warming,
the summer ice age is on the decrease, and the ice microstructure and IOPs change accordingly, leading to an overall higher
ice transmittance. Furthermore, the transmitted solar energy affects the temperature of the upper ocean and results in further
melting of the bottom of sea ice (Timmermans, 2015). Along with the melting of ice, gas bubbles, and brine pockets change
simultaneously (Light et al., 2004), which affects the IOPs of ice in turn. Consequently, the sea ice is expected to become
thinner and more porous than before. This process has been seldom considered in previous studies. Related studies generally
regarded the surface properties and thickness of the ice as predictors for light transmittance (Katlein et al., 2015; Perovich et
al., 2020). The microstructure and morphological parameters of sea ice (e.g., thickness, extent, etc.) may together influence
the melting processes of Arctic sea ice.
For safe field observations, the ice core data used in this study were all sampled in large and thick floes. Therefore,
variations in the microstructure of the ice in marginal zones or under melt ponds cannot be addressed by this study. Light et al.
(2015) reported that the differences in the $\sigma$ between the IL of ponded first-year ice and multiyear ice were larger than those
between bare first-year ice and multiyear ice. Therefore, the changes in the IOPs of the marginal ice zone were expected to be
more obvious than those found in the present results because the ice in marginal zones is more likely young and ponded (Rigor
and Wallace, 2004; Zhang et al., 2018). Furthermore, the same changes in the ice microstructure have more effects on the $T_B$
of thin sea ice (Section 3.4). Marginal ice zones, comprising 16.4% of the entire ice area, contributed 39.3% of the extra-
transmitted solar energy due to the changing ice microstructure from 2008 to 2016 (Figure 8). Both processes promote an
increase of transmitted flux through sea ice and ice bottom melting in marginal ice zones. Arndt & Nicolaus (2014) quantified
light transmittance through the sea ice into the ocean for all seasons as a function of variable sea ice types. The mean annual
trend was 1.5% per year, which mainly depended on the timing of melt onset. If the variations in the microstructure of bare
and ponded ice are taken into consideration, this trend is expected to increase. We suggest that future ice observations and
models should pay more attention to variations in the ice age, microstructure, and their effects, especially in marginal ice zones.
We want to emphasize the Arctic basin-scale analysis is a highly idealized investigation. To obtain a real distribution of
the transmitted solar radiation through sea ice in the Arctic basin scale in the summer is far more complicated and would
require a massive amount of ice core sampling collected simultaneously in various parts of the Arctic Ocean. Such field
expeditions cannot be arranged anytime soon in the future. We intend to provide one possible scenario of IOPs. We call for
further strengthening international collaborations to make possible a better understanding of the Arctic IOPs distribution.
**5    Conclusions**
This is the first study to link the ice microstructure, IOPs, and AOPs at interannual scales. Based on ice cores sampled
during the CHINARE expeditions (2008–2016), the variations in the IOPs of Arctic sea ice in summer due to the changing
microstructure of ice were modeled according to structural-optical theory. Variations in the AOPs and solar flux distribution
due to the changing IOPs in the summer Arctic were also estimated. Clear variations in the microstructure and IOPs of each
year (Figure 5) enabled us to construct a quantitative view of changes that the Arctic sea ice interior underwent in these years.
As a result of our study, there were no significant variations in the microstructure and IOPs of ice TL. This is related to
the stable melt days in study years. Because $\sigma$ of the upper layers (TL and DL) mainly control the albedo of bare ice, the
variations in $\alpha_B$ between years were relatively small. Meanwhile, variations in the microstructure and IOPs of IL were
significant. These variations consist mainly of interannual factors and minor spatial factors. After excluding the effects of
spatial variations, we found these interannual variations in $\sigma$ of ice IL were highly related to the changing ice ages. That's to
say, the ice age largely manifested in the ice microstructure of the IL. The changing $\sigma$ of ice IL affects the ice transmittance
clearly. Furthermore, the same changes in the ice IOPs had more effects on the transmittance of the thin ice in marginal ice
zones.
Previous studies paid more attention to changing transmittance due to declining ice thickness. The present findings
demonstrated that the changing IOPs of interior ice derived from the ice microstructure could also alter the partitioning of solar
radiation in sea ice by itself. With continued Arctic warming, summer ice will become younger and more porous than before,
leading to more light reaching the upper ocean. This reminds us to pay more attention to the variations in the IOPs of interior
ice, especially ice with different ages.

*Acknowledgments.* We would like to thank Handing editor Dr. Marie Dumont and 4 anonymous reviewers. Their criticism and constructive comments helped to improve this manuscript significantly. We are grateful to the NSIDC, Alfred Wegener Institute, and ECMWF for providing the sea ice and radiation data. This work was financially supported by the National Key Research and Development Program of China (Grant number 2018YFA0605901), the National Natural Science Foundation of China (Grant numbers 41922045, 41906198, 41976219, and 41876213), and the Academy of Finland (Grant numbers 333889, 325363, and 317999). We also wish to acknowledge the crews of the R/V Xuelong for their fieldwork during CHINARE.

*Author contributions.* MY carried out the estimations and wrote the paper. RL, BL, and QW provided the ice core data. All coauthors discussed the results and edited the manuscript.

*Data Availability Statement.* The sea ice and radiation data are available at https://doi.org/10.5067/MPYG15WAA4WX; https://data.meereisportal.de/gallery/index_new.php?lang=en_US&ice-type=extent&active-tab1=measurement&active-tab2=thickness; https://cds.climate.copernicus.eu/cdsapp#!/dataset/reanalysis-era5-single-levels-monthly-means?tab=form. The ice cores data applied in this work can be accessed in Wang et al. (2020).

*Competing interests.* One of the co-authors is an editor of The Cryosphere. The peer-review process was guided by an independent editor, and the authors have also no other competing interests to declare.

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
