# Peer review of "Modeled variations of the inherent optical properties of summer"

_EGUsphere, 2022_

## Author Comment (AC1)

Dear Reviewer:

Thank you for your comments concerning our manuscript (ID: egusphere-2022-552). Those comments were very helpful for revising and improving this manuscript, as well as for providing important guidance to our study. We have considered the comments carefully and will make enough changes to the manuscript. The responses to the reviewer's comments are provided in blue, as follows.

This manuscript describes a study using sea ice microstructural property observations recorded over a broad region in the Arctic Pacific sector during the interval 2008 – 2016 to compute changes in the inherent optical properties of the observed ice. This paper takes air volume and brine volume observed by Wang et al. (2020) as the basis for computing inherent optical properties (scattering coefficient, absorption coefficient, and scattering phase function asymmetry parameter) and apparent optical properties (albedo and transmittance) for sea ice.

The text and figures are clear as presented. I do have major concerns with the method and the conclusions that were reached. There is a general lack of rigorous statistical treatment applied to this dataset. To do this study of interannual variability correctly, it is necessary to first establish the (regional) spatial and temporal variability in a single year. The variability in microstructure properties is affected by temperature and number of melt days, but also potentially by absorption of shortwave radiation, melt water flushing, synoptic weather (e.g., rain events), surface vapor condensation, surface melt pooling, and other factors. Many of these processes would be expected to drive significant spatial and temporal variability in the brine and gas volumes in sea ice, especially in the uppermost portions of the ice cover. The spatial and temporal sampling are not adequate to draw the conclusion that the brine and gas volumes have changed in response to spatially large and temporally long changes in climate.

We have supplied more statistical treatment to the ice core data. Then, the effects of spatial and temporal variability on the results can be reduced. The conclusion has been revised accordingly. First, to reduce the impact of temporal variations in the ice cores on the ice microstructure, we preprocessed the ice core data. The ice cores in each year were allocated different weights according to their sampling date. Subsequently, Figures 2–8 have been updated. It is a big challenge to make clear the effects of shortwave radiation, flushing, rain events, etc. on ice properties according to the

available literature. Whereas, the mean sampling date of ice cores was Aug. 18 ± 9 days (described in Section 4.2). According to the previous observations, ice surface melt was most intensive in July, and relatively stable in August (Perovich et al. 2003; Nicolaus et al. 2021). Meanwhile, it can be seen from the melt onset data that the melting days of the sampling sites were similar (58 ± 7 days). In other words, although the radiation, flushing process, etc., affected the seasonal variations in the microstructure of ice, their combined effects did not introduce many variations to the melting days of ice. It could be expected that the brine and gas volumes in sea ice will not change obviously over the course of several days.

As for the spatial variations in the ice cores, more discussion has been supplied in Section 4.2. We have quantitatively analyzed the effects of spatial variations on the interannual variations through the propagation law of error. Some conclusions have also been amended accordingly. As we know, it is difficult for field observations to avoid the effects of spatial variations. Therefore, related studies have generally ignored the effects of sampling locations on the statistics (e.g. Carnat et al. 2013; Katlein et al. 2019; Frantz et al. 2019). In the present study, Fig. 10 shows the inherent optical properties of the ice cores in different latitude zones. It can be seen that there were indeed spatial variations in the ice properties. However, we noted that the spatial variations were relatively smaller than the interannual variations. In other words, interannual factors played a more important role in the ice properties than spatial factors. This was what we have sought to express clearly in Section 4.2.

Lines 15 -17 (abstract) illustrate my point: "Compared with 2008, the volume fraction of gas bubbles in the top layer of sea ice in 2016 increased by 7.5%, and decreased by 50.3% in the interior layer. Meanwhile, the volume fraction of brine pockets increased clearly in the study years." With no knowledge of the spatial or temporal variability of these properties within a single region / year, attribution of their interannual variability is unfounded.

The abstract has been rewritten. Information about the effects of the spatial and temporal variations on the interannual variations has been supplied. We have quantitatively shown the roles of the interannual and spatial variations on the changing microstructure and optical properties of ice.

The temporal variability question here may be tied to the sampling period. Line 59-60 reads "Almost all cores were sampled in August, when the ice had started to melt." I would argue that data taken in August likely exhibit very strong short-term temporal variability. By August, the ice surface has likely been melting (losing mass) for at least a month. It is also possible that by August the surface melt has ceased. The brine and gas volumes may thus be changing quickly, and not monotonically at this summer/autumn transition time. It is possible that the sampling was carried out without spatial or temporal biases, but the authors have not presented a convincing statement that this is true.

As stated in the reply to another comment, we preprocessed the ice core data to reduce the effect of the temporal variations in the ice cores on the statistical results. The weights of the ice cores sampled at early or later dates have been reduced. Meanwhile, according to the melt data from NASA, the ice cores in the present manuscript were all sampled during the late melting season. As such, the sea ice had been melting for a while (~58 days) and had not yet begun to freeze (it needed another ~15 days). According to previous observations, the ice surface melt rate in August was only ~1/10 of that in July (Perovich et al. 2003; Nicolaus et al. 2021). Therefore, short-term temporal variability was expected not to affect ice obviously. This information has been supplied in the revised manuscript.

Line 143 "There were clear increases in the Vb of all three ice layers (Figure 3b), which implied dramatic variations in the permeability of summer sea ice." There is no discussion of how permeability is measured or modeled.

Yes, the permeability of ice was not the main target of the present manuscript. This sentence has been rewritten in the revised manuscript.

Line 145: "From 2008 to 2016, the increase in the IL was clearest." This is a qualitative statement and contains no robust statistical assessment.

Statistical descriptions of variations has been supplied in the revised manuscript.

What physics drive changes in sea ice scattering coefficient? Temperature is certainly a primary driver, at least initially. But it is by no means the only driver. Once the ice surface is melting its temperature changes little.

For now, we have little quantitative knowledge of the seasonal progression of the sea ice scattering coefficient or microstructure and its influencing factors. In 2008, Light et al. first showed some evolution of IOPs during the course of the summer for the multiyear ice observed at SHEBA, but did not discuss their influencing factors much. The reason we analyzed the relationships between temperature and the ice scattering coefficient was because the former was an important parameter in sea ice models. Furthermore, temperature was easy to measure. If empirical or semi-empirical relationships between temperature and the scattering coefficient could be determined, it would be useful for related studies and models to set ice parameters.

Lines 156 – 158 "2). Although the Vb values of the ice cores increased clearly with depth, they did not enhance the scattering capacity of ice. The reason for this was that the refractive indices of brine pockets and pure ice are close (Smith and Baker, 1981; Grenfell and Perovich, 1981)." It is certainly true that the refractive indices of brine and ice are close, but even small changes affect scattering.

What we wished to express was that the effects of the changing brine pocket volume was covered by the changing gas bubble volume. This sentence has been rewritten to reduce the previous ambiguity.

Section 3.3. Are the reported AOPs observations? Or are they calculated with a radiative transfer model? Caption for Fig. 6 says "estimated", so I am left to infer these are calculated, not observed. It would be interesting if there were a comparison between these calculated values and observed values.

Yes, they were estimated. Some descriptions will be supplied here. Similar parameterizations have been widely used to link ice microstructure with optical properties and have been verified by extensive observations (e.g. Taskjelle et al., 2017). Radiative transfer models are also commonly used, whose accuracies are widely accepted.

Line 231 – 233 asserts: "Meanwhile,Ea decreased from 15 W m-2 in 2008 to 13.8 W m-2 in 2016. As the decrease in ice volume from 2008 to 2016 was 32.2%, the solar energy absorbed by a unit volume of sea ice increased by 35.7% on the Arctic scale." This would be an interesting result if it was based on rigorous assessment. It is difficult to discern however whether it is rather based on propagated error.

Similar results have been observed in a related study. Section 4.2 of the present manuscript showed that the ice ages of these ice cores were different. Light et al. (2015) showed that the thickness of first-year ice was less by 13.3% than multiyear ice (1.3 m vs. 1.5 m, respectively). Whereas, the radiation absorbed by the former was less by 2% than the latter. In other words, the solar energy absorbed by a unit volume of first-year ice was greater than multiyear ice by 12.5%.

Lines 300 – 305: "Extensive measurements of the IOPs of Arctic sea ice have been carried out, and some authors have noticed the seasonal variations of the ice microstructure and IOPs (e.g., Light et al., 2008; Frantz et al., 2019; Katlein et al., 2021). However, interannual variations in sea ice IOPs are still not clear, although such changes in sea ice extent, thickness, and age are evident. A lack of continuous IOP measurements is the primary reason. Compared with previous observations, the ice core data in the present study were more appropriate for interannual analyses of the IOPs of ice because of their long time span and consistencies in the sampling method, seasons, and sea areas." Yes, I agree with this statement. I also agree this data set is "more appropriate". But, "more appropriate" still needs to be handled carefully. I don't find it appropriate to assume that because it is "more appropriate" that it is appropriate enough.

Yes, we agree with you. We have supplied more information about the spatial and temporal variations of the ice properties. To reduce the temporal variations, the weights of the ice cores sampled at early and late dates have been reduced. Quantifying the effects from spatial variations is a big challenge because little quantitative knowledge is known about them. Instead, we have quantitatively analyzed their effects on the interannual changes. Although the present ice core data set did not form a strict time series in the classical sense, it could be used to derive a qualitative picture of the changing ice microstructure. After making the temporal and spatial variations clear, the

interannual variations then become clear, and we have sought to unambiguously convey our results in the revised manuscript.

Lines 316 – 317: "For σ, there were no clear changes in the TL. This demonstrated that the variations of σ in the TL largely resulted from interannual factors." I completely agree. But there is no elaboration on what these interannual factors could be. Rain/snow? Ice dynamics? Length and intensity of melt season?

In the following part of Section 4.2, we discussed the interannual variations in the melting days, temperature, surface radiation, and ice age. They were all important factors related to the development of ice properties. We found that melt days and radiation in the study years were relatively stable, while the air temperature and ice age clearly varied. Thus, Figure 11 mainly discusses the effects of these two latter parameters on the ice optical properties. More discussion has been supplied in this section in the revised manuscript.

Lines 317 – 318: "With an increase of latitude, the σ of the IL tended to increase." Yes, it would be expected that the ice at lower latitude is generally warmer earlier in the season. This internal warming would be expected to lead to increased brine inclusion size and connectivity. This connectivity would naturally lead to brine drainage, and reduced scattering coefficient. This seems like a useful, justifiable result, but I don't believe this is the point being made here.

Yes, what we wished to express in this paragraph is there were, inevitably, spatial variations in the ice cores. Furthermore, the spatial variations were relatively less than the interannual variations. Following this, a discussion regarding the reason for the interannual variations has been provided in the text. We have also reorganized this part to make it clearer than before.

Lines 325 – 326: "The amount of surface radiation during the study years was also similar (Laliberté et al., 2021)." This is a very sweeping generalization. I would expect the details of this study to be quite sensitive to short time scale variations within this generalized picture, and for the ice state to respond to these variations.

The mean surface radiation in the study years was 99.4 ± 6 W, which was determined from the reanalysis data (ECMWF). Definitely, short-time scale variations in radiation affected the ice properties. In this section, we have paid attention to the interannual variations of ice optical properties and their reasons. After checking the relationships between the ice optical properties and potential affecting factors, we found that the interannual variations in the surface radiation could hardly explain the changing ice optical properties. Indeed, there were some more important factors resulting in these variations. We have supplied more descriptions of the variations in other climate factors in this section, and have discussed the relationship between these factors and the ice properties.

Figure 11(a): I would expect T_air to have synoptic (temporal and spatial) variability. I would expect TL scattering coefficient to be sensitive to integrated solar radiation and surface vapor deposition. I think the correlation implied by this figure (as stated in Lines 348 – 349 "In summary, the differences in the IOPs of the ice cores were related to interannual variations in the air temperature and ice age" is misleading.

Yes, there were temporal and spatial variations in air temperature. We would like to emphasize that the increasing temperature in Figure 11(a) was not an exception but a general circumstance in the Arctic during 2008–2016 (Collow et al., 2020). This could also be seen in the reanalysis data, where the mean air temperature in the summer of the study area has been increasing gradually (0.12 ℃/year, correlation coefficient r = 0.84). This trend agreed well with our observed temperature measurements (0.14 ℃/year, r = 0.84). However, there did not appear to be a clear trend in the solar radiation (r < 0.5), and the correlations between radiation and the ice optical properties were not clear. So, although solar radiation and vapor deposition could affect the seasonal variations in ice clearly, we thought that the air temperature played a more important role in the interannual variation in ice. We have supplied more discussion about these factors to make our view clear.

---

## Author Comment (AC3)

Dear Dr. Marie Dumont:

Thanks a lot for your letter concerning our manuscript (ID: egusphere-2022-552). I'd like to provide some detailed changes in the results, discussion, and conclusion performed in the manuscript in response to the main concern of the reviewer.

In the method section, we added some information about the dataset and the method to preprocess the ice core data. After the preprocessing, temporal uncertainties of ice properties in each year were reduced. In the revised manuscript, about half of the text in Section 2.1 was newly provided.

In the result section, all figures (Figures 2-8) have been updated. On the one hand, the data after preprocessing are not the same as the original version. On the other hand, the color of lines was adapted to avoid indistinction for color-blind readers. Analyses in this section were also revised accordingly. The treatment of uncertainty was supplied as a part of all analyses done in the manuscript. About 40% of analyses in Sections 3.1, 3.2, and 3.3 was revised. After the analysis of variance, we found that the variation in the top layer of ice between years was statistically insignificant. Meanwhile, the interannual variations in the interior layer were significant. These were also the main changes in the conclusion.

Section 4.2 of the discussion was almost completely rewritten (three pages). Figure 11 was also remade. We discussed the potential spatial and temporal variability in ice cores in response to the main concern of the reviewer. We found that the shortwave radiation in the study years was nearly constant (99.4 ± 6 W). Furthermore, variation in melting days of ice sampling sites between years was statistically insignificant (ANOVA, $P > 0.2$). Previous observations also demonstrated that ice surface melt was relatively stable in August (e.g. Perovich et al., 2003; Nicolaus et al., 2021). Therefore, we think the effects of temporal variations on the ice surface were relatively small and can be ignored. As for the ice interior layer, its variations were significant. Previous observations have reported that properties of the interior layer were nearly constant in the melt season (e.g. Light et al., 2008; Frantz et al., 2019). So, the variations in the ice interior layer did not result from the temporal variability.

The spatial differences in ice cores have been provided in the original manuscript. In the last version, we found there were no clear changes in the optical properties of the top layer in different latitude zones. Furthermore, the scattering coefficient of the ice interior layer from the low-latitude zone was relatively small. The reviewer believed these were "useful, justifiable results". Combined with the analysis in Section 3.2, we found the top layer of ice cores has no significant temporal, spatial, or interannual

variations, and the variations in the interior layer consisted of interannual and spatial variations. Then, we further quantitatively discussed the effects of spatial variations on the whole variations of the interior layer through the propagation law of variation (whole variations$^2$ = interannual variations$^2$ + spatial variations$^2$). Some ice cores in 2014 and 2016 were sampled in the low-latitude zone. All ice cores in 2008, 2012, and 2014 were sampled in mid- or high-latitude zones. According to the differences between ice cores from different years (Figure 3) and different latitude zones (Figure 10), we correct the scattering coefficient of the interior layer in 2014 from 176 m$^{-1}$ to 182 m$^{-1}$. That's to say, the interannual variations were larger than the whole variations by 6 m$^{-1}$. The value of 2016 was also corrected accordingly. Then, variations between the corrected scattering coefficient of the interior layer could be regarded as the result of the interannual factors.

In the revised conclusion, we emphasized that there are no statistically significant spatial, temporal, and interannual variations in the ice top layer. Meanwhile, variations in the interior layer were significant and mostly resulted from the different ice ages.

We hope these descriptions of changes performed in the manuscript are helpful for you to proceed with a decision on the manuscript. Thanks a lot.

---

## Author Response (AR1)

Dear Reviewer:

Thank you for your comments concerning our manuscript (ID: egusphere-2022-552). Those comments were very helpful for revising and improving this manuscript, as well as for providing important guidance to our study. We have considered the comments carefully and have made enough changes to the manuscript. The responses to the reviewer's comments are provided in blue, as follows.

This manuscript describes a study using sea ice microstructural property observations recorded over a broad region in the Arctic Pacific sector during the interval 2008 – 2016 to compute changes in the inherent optical properties of the observed ice. This paper takes air volume and brine volume observed by Wang et al. (2020) as the basis for computing inherent optical properties (scattering coefficient, absorption coefficient, and scattering phase function asymmetry parameter) and apparent optical properties (albedo and transmittance) for sea ice.

The text and figures are clear as presented. I do have major concerns with the method and the conclusions that were reached. There is a general lack of rigorous statistical treatment applied to this dataset. To do this study of interannual variability correctly, it is necessary to first establish the (regional) spatial and temporal variability in a single year. The variability in microstructure properties is affected by temperature and number of melt days, but also potentially by absorption of shortwave radiation, melt water flushing, synoptic weather (e.g., rain events), surface vapor condensation, surface melt pooling, and other factors. Many of these processes would be expected to drive significant spatial and temporal variability in the brine and gas volumes in sea ice, especially in the uppermost portions of the ice cover. The spatial and temporal sampling are not adequate to draw the conclusion that the brine and gas volumes have changed in response to spatially large and temporally long changes in climate.

We agree with your concerns about the statistical treatment applied to this ice data. It is "perfect" if the sampling was carried out without spatial or temporal biases. Whereas, this "perfect" data is nonexistent in the present literature. If the MOSAiC can carry out for 5-10 years, it may produce this "perfect" data. But this is extremely difficult. For now, ice core data from the CHINARE is the only dataset with spatial and temporal continuity. In the revised manuscript, we do substantial revisions to the

method, results, discussion and conclusion to make them more rigorous than before. More statistical treatments have been supplied to the ice core data. Then, the effects of spatial and temporal variability on the results can be reduced or clear. The conclusion has been revised accordingly. We consider the presented ice core data to be the best possible estimate of the potential interannual variations at this time while acknowledging that further improvements of the data are needed. It is conceivable that more available ice data in the future might yield more rigorous and different results, although such differences are likely to be insignificant for the conclusions presented here.

Firstly, for the seasonal variations in ice microstructure, Macfarlane et al. (2023) found that the surface layer microstructure of melting ice showed no temporal changes in the entire July using X-ray micro-computed tomography. Furthermore, Perovich (2003) and Nicolaus et al. (2021) demonstrated that ice surface melt in August was weaker than in July. Meanwhile, the scattering coefficients of the interior ice layer are relatively constant during the entire melt season (Light et al., 2008). So, it is expected that the effects of temporal variations on the microstructure and IOPs of the surface and interior ice in August were relatively small.

To further reduce the potential impact of temporal variations in the ice cores on the ice microstructure and IOPs, we preprocessed the ice core data. The ice cores in each year were allocated different weights according to their sampling date (L76-81). After the preprocessing, the deviation of melting days in a single year was reduced by ~50.5%. Furthermore, the treatment of uncertainty was supplied as a part of all analyses done in the manuscript (Section 3). After the analysis of variance (ANOVA), we found that the variation in the top layer of ice between years was statistically insignificant. Meanwhile, the interannual variations in the interior layer were significant. These were also the main changes in the conclusion. Subsequently, Figures 2–8 have been updated.

The variability in ice surface properties during melting is directly controlled by the number of melt days. The melt days are affected by longwave radiation, water vapor, air temperature, and other factors (Persson, 2012; Mortin et al., 2016; Crawford et al., 2018). We found the amount of surface downward longwave, total column vertically-integrated water vapor during the study years have no clear variations or statistically significant trends (Fig. 11a). As a result, the melting days of the sampling

sites were similar (59 ± 7 days). Their variations between years were statistically insignificant (ANOVA, P > 0.1). Meanwhile, variations in surface ice microstructure are also statistically insignificant. In other words, although the radiation, flushing process, etc., may affect the seasonal variations in the microstructure of ice, their combined effects did not introduce many variations to the melting days of ice in different years. It could be expected that the brine and gas volumes in sea ice will not change obviously over the study years.

Secondly, as for the spatial variations in the ice cores, more discussion has been supplied in Section 4.2. Some conclusions have also been amended accordingly. As we know, it is difficult for field observations to avoid the effects of spatial variations. Therefore, related studies have generally ignored the effects of sampling locations on the statistics (e.g. Carnat et al. 2013; Katlein et al. 2019; Frantz et al. 2019). In the present study, Fig. 10 shows the inherent optical properties of the ice cores in different latitude zones. It can be seen from Figure 10a, there were no clear changes in the mean $\sigma$ of TL in different latitude zones. Meanwhile, the scattering coefficient of the ice interior layer from the low-latitude zone was relatively small. According to Light et al.(2008), the $\sigma$ of the IL is relatively constant during the entire melt season. That's to say, the variations in the ice interior layer didn't result from temporal factors. So, the whole variations in the interior layer consisted of interannual and spatial variations. Then, we further quantitatively discussed the effects of spatial variations on the whole variations of the interior layer through the propagation law of variation. Some ice cores in 2014 and 2016 were sampled in the low-latitude zone. All ice cores in 2008, 2012, and 2014 were sampled in mid- or high-latitude zones. According to the differences between ice cores from different years (Figure 3) and different latitude zones (Figure 10), we correct the scattering coefficient of the interior layer in 2014 from 176 m$^{-1}$ to 182 m$^{-1}$. That's to say, if the ice cores in 2014 were sampled in the mid- or high-latitude, their scattering coefficient of the interior layer will increase by 6 m$^{-1}$. The value of 2016 was also corrected accordingly. Then, variations between the corrected scattering coefficient of the interior layer could be regarded as the result of the interannual factors.

Lines 15 -17 (abstract) illustrate my point: "Compared with 2008, the volume fraction of gas bubbles in the top layer of sea ice in 2016 increased by 7.5%, and

decreased by 50.3% in the interior layer. Meanwhile, the volume fraction of brine pockets increased clearly in the study years." With no knowledge of the spatial or temporal variability of these properties within a single region / year, attribution of their interannual variability is unfounded.

This part of the abstract has been rewritten. Information about the effects of spatial and temporal factors on interannual variations has been analyzed. We have supplied the analysis of variance for the whole manuscript, we found that the variation in the top layer of ice between years was statistically insignificant. This sentence has been revised to "The variations in $V_a$ of the ice top layer were not significant and $V_a$ of the ice interior layer was significant. Compared with 2008, the mean $V_a$ of interior ice in 2016 decreased by 9.1%. Meanwhile, the volume fraction of brine pockets increased clearly in the study years."(L15-17)

The temporal variability question here may be tied to the sampling period. Line 59-60 reads "Almost all cores were sampled in August, when the ice had started to melt." I would argue that data taken in August likely exhibit very strong short-term temporal variability. By August, the ice surface has likely been melting (losing mass) for at least a month. It is also possible that by August the surface melt has ceased. The brine and gas volumes may thus be changing quickly, and not monotonically at this summer/autumn transition time. It is possible that the sampling was carried out without spatial or temporal biases, but the authors have not presented a convincing statement that this is true.

In the latest MOSAiC observations, Macfarlane et al. (2023) found that the surface scattering layer microstructure of melting ice remained constant for the entire July. Furthermore, the ice surface melt rate in August was only ~1/10 of that in July (Perovich et al. 2003; Nicolaus et al. 2021). Therefore, short-term temporal variability was expected not to affect ice microstructure obviously. Furthermore, as stated in the reply to another comment, we preprocessed the ice core data to reduce the effect of the temporal variations in the ice cores on the statistical results. The weights of the ice cores sampled at early or later dates have been reduced. Meanwhile, according to the melt data from NASA, the ice cores in the present manuscript were all sampled during the late melting season. As such, the sea ice had been melting for a while (~58 days) and

had not yet begun to freeze (it needed another ~15 days). This information has been supplied in the revised manuscript.

Line 143 "There were clear increases in the Vb of all three ice layers (Figure 3b), which implied dramatic variations in the permeability of summer sea ice." There is no discussion of how permeability is measured or modeled.

Yes, the permeability of ice was not the main target of the present manuscript. This sentence has been rewritten in the revised manuscript (L172).

Line 145: "From 2008 to 2016, the increase in the IL was clearest." This is a qualitative statement and contains no robust statistical assessment.

Statistical descriptions of variations have been supplied in the whole revised manuscript. The revised sentence is "the increases of mean $V_b$ in the IL were statistically significant (r = 0.84, P = 0.07; ANOVA, P < 0.01)."(L173)

What physics drives changes in sea ice scattering coefficient? Temperature is certainly a primary driver, at least initially. But it is by no means the only driver. Once the ice surface is melting its temperature changes little.

For now, we have little quantitative knowledge of the seasonal progression of the sea ice scattering coefficient or microstructure and its influencing factors (Light et al., 2015). In 2008, Light et al. first showed some evolution of IOPs during the summer for the multiyear ice observed at SHEBA but did not discuss their influencing factors much. In Section 4.2 of the revised manuscript, we have supplied the discussion on longwave radiation and vapor during the study years (L369-373). As a result, we didn't find these factors have clear variations. Meanwhile, although the air temperature in the study years was increasing, it seems not the predominant affecting factor.

Lines 156 – 158 "2). Although the Vb values of the ice cores increased clearly with depth, they did not enhance the scattering capacity of ice. The reason for this was that the refractive indices of brine pockets and pure ice are close (Smith and Baker,

1981; Grenfell and Perovich, 1981)." It is certainly true that the refractive indices of brine and ice are close, but even small changes affect scattering.

What we wished to express was that the effects of the increasing brine pocket volume were covered by the decreasing gas bubble volume. This sentence has been rewritten to reduce the previous ambiguity (L186-189).

Section 3.3. Are the reported AOPs observations? Or are they calculated with a radiative transfer model? Caption for Fig. 6 says "estimated", so I am left to infer these are calculated, not observed. It would be interesting if there were a comparison between these calculated values and observed values.

Yes, they were estimated. Some descriptions have been supplied (L222-223). Similar parameterizations have been widely used to link ice microstructure with optical properties and have been verified by extensive observations (e.g. Taskjelle et al., 2017). Radiative transfer models are also commonly used, whose accuracies are widely accepted.

Line 231 – 233 asserts: "Meanwhile, Ea decreased from 15 W m-2 in 2008 to 13.8 W m-2 in 2016. As the decrease in ice volume from 2008 to 2016 was 32.2%, the solar energy absorbed by a unit volume of sea ice increased by 35.7% on the Arctic scale." This would be an interesting result if it was based on rigorous assessment. It is difficult to discern however whether it is rather based on propagated error.

Similar results have been observed in a related study. Section 4.2 of the present manuscript demonstrated that the ice ages of these ice cores were different. Light et al. (2015) showed that the thickness of first-year ice was less by 13.3% than multiyear ice (1.3 m vs. 1.5 m, respectively). Whereas, the radiation absorbed by the former was less by 2% than the latter. In other words, the solar energy absorbed by a unit volume of first-year ice was greater than multiyear ice by 12.5%. This information has been supplied in L239-242.

Lines 300 – 305: "Extensive measurements of the IOPs of Arctic sea ice have been carried out, and some authors have noticed the seasonal variations of the ice microstructure and IOPs (e.g., Light et al., 2008; Frantz et al., 2019; Katlein et al., 2021). However, interannual variations in sea ice IOPs are still not clear, although such changes in sea ice extent, thickness, and age are evident. A lack of continuous IOP measurements is the primary reason. Compared with previous observations, the ice core data in the present study were more appropriate for interannual analyses of the IOPs of ice because of their long time span and consistencies in the sampling method, seasons, and sea areas." Yes, I agree with this statement. I also agree this data set is "more appropriate". But, "more appropriate" still needs to be handled carefully. I don't find it appropriate to assume that because it is "more appropriate" that it is appropriate enough.

Yes, we agree with you. As replied in another comment, we have supplied more information about the spatial and temporal variations of the ice properties. According to published studies, there are no clear temporal variations in ice surface microstructure and interior ice IOPs (Macfarlane et al. 2023; Light et al., 2008). In the revised manuscript. the weights of the ice cores sampled at early and late dates have been reduced to further reduce the temporal variations. Quantifying the effects of spatial variations is a big challenge because little quantitative knowledge is known about them. Instead, we have quantitatively analyzed their effects on the interannual changes. Although the present ice core data set did not form a strict time series in the classical sense, it could be used to derive a qualitative picture of the changing ice microstructure. We acknowledge that the present data products are not perfect while considering the presented ice core data is the best possible estimate of the interannual variations after making the temporal and spatial variations clear.

Lines 316 – 317: "For $\sigma$, there were no clear changes in the TL. This demonstrated that the variations of $\sigma$ in the TL largely resulted from interannual factors." I completely agree. But there is no elaboration on what these interannual factors could be. Rain/snow? Ice dynamics? Length and intensity of melt season?

In Section 4.2, we discussed the interannual variations in the melting days, temperature, longwave radiation, vapor, and ice age. They were all important factors

related to the development of ice properties. Figure 11 detailly shows the variations in these factors. We found that the melt days, vapor, and radiation in the study years were relatively stable, while the air temperature and ice age varied. Furthermore, the increasing air temperature seems to have little effect on the ice microstructure. Section 4.2 has been revised substantially. More details can be found in the revised manuscript.

Lines 317 – 318: "With an increase of latitude, the σ of the IL tended to increase." Yes, it would be expected that the ice at lower latitudes is generally warmer earlier in the season. This internal warming would be expected to lead to increased brine inclusion size and connectivity. This connectivity would naturally lead to brine drainage and reduced scattering coefficient. This seems like a useful, justifiable result, but I don't believe this is the point being made here.

We have reorganized the discussions of Section 4.2. The first half part (L367-394) discussed the spatial, temporal, and interannual variations in the TL of ice. The discussion about IL was shown in L400-432. The mentioned sentence was moved to the latter half part. After establishing the difference of IL in different latitude zones (Fig.10), we further quantitatively discussed the effects of spatial variations on the whole variations of the interior layer through the propagation law of variation.

Lines 325 – 326: "The amount of surface radiation during the study years was also similar (Laliberté et al., 2021)." This is a very sweeping generalization. I would expect the details of this study to be quite sensitive to short time scale variations within this generalized picture, and for the ice state to respond to these variations.

The amount of surface downward longwave radiation in the study years was 300.2 ± 4.0 W/m$^2$, which was got from the reanalysis data (ECMWF). The standard variations in single years were also small (1.1 W/m$^2$). So, there are no clear differences in radiation between years or in a single year. Definitely, short-time scale variations in radiation affected the ice properties. Whereas, these effects were not clear enough. As the observations of Macfarlane et al. (2023) in the latest MOSAiC, the surface layer microstructure of melting ice shows no clear temporal change for a month. Therefore, it is expected that the effects of temporal variations on the microstructure and IOPs of

the present ice core surface were relatively small. We have supplied more descriptions of the variations in other climate factors in this section (Fig.11), and have discussed the relationship between these factors and the ice properties.

Figure 11(a): I would expect T_air to have synoptic (temporal and spatial) variability. I would expect TL scattering coefficient to be sensitive to integrated solar radiation and surface vapor deposition. I think the correlation implied by this figure (as stated in Lines 348 – 349 "In summary, the differences in the IOPs of the ice cores were related to interannual variations in the air temperature and ice age" is misleading.

We have rewritten this part to make our view clear. Yes, there were temporal and spatial variations in air temperature. Whereas, the standard variations of temperature in single years were relatively small (0.3℃). Meanwhile, we would like to emphasize that the increasing temperature in Figure 11(a) was not an exception but a general circumstance in the Arctic during 2008–2016 (Collow et al., 2020). This could also be seen in the reanalysis data, where the mean air temperature in the summer of the study area has been increasing gradually (0.12℃/year, correlation coefficient r = 0.84). This trend agreed well with our observed temperature measurements (0.14℃/year, r = 0.84).

It is reasonable that the TL scattering coefficient is sensitive to radiation and surface vapor deposition. Whereas, there did not appear to be clear variations in the longwave radiation or column vertically-integrated water vapor (see L369-373, Fig. 11) between different years. The difference between different cores in a single year is also not clear (the coefficient of variations were 0.003 and 0.02 respectively). For now, we have little quantitative knowledge of the seasonal progression of the sea ice scattering coefficient. Whereas, the latest observation in MOSAiC shows the ice surface maintains a consistent microstructural profile throughout the melt season. So, it is expected that the scattering coefficient of TL also has no clear seasonal variations. A detailed discussion can be found in the revised Section 4.2.

The manuscript describes a study of Arctic sea ice cores, collected from a series of research cruises, to determine optical properties of sea ice and their general variability across 2008-2016. I found the manuscript mostly quite well written, with a pleasingly broad range of cited prior literature. The results emerged from a long period of arduous fieldwork and present a consistent message on the evolution of Arctic sea ice when compared to other fieldwork and remote sensing studies on the topic. Some concerns remain, mainly on the generalization of results to pan-Arctic scale and some aspects of the presentation. If the authors can address them, reaching publishable quality should be possible.

Major comments:

1. Sections 2.3 and 4 were difficult to follow in terms on what is actually done to arrive at Fig 7, and what the results actually represent. There are several concerns here:

o   How specifically is the interannual albedo of ice calculated?

The ice optical properties are calculated through the radiation transfer model (Section 2.2). The input parameters include the grided ice thickness (from remote sensing data, see Section 2.3) and inherent optical properties (from the ice cores data). We have now introduced the process more clearly in the method section (L137-139).

o   What does it represent – the (mean?) broadband albedo of pure ice derived from all ice cores of each year?

Here, the observed ice microstructure from ice cores was employed to estimate ice optical properties. The impurity and ice surface properties (such as a snow layer or melt ponds) were not considered here. Thus, these results are based on an ideal situation, and could not be fully representative of the real situation. This is because what we focused on was the effects of the ice microstructure on their optical properties. We have supplied some relevant information in the related section (L222-225).

o   Is it equivalent to black-sky albedo, where atmospheric conditions would not matter? The text around lines 93-101 suggest that white-sky albedo is being derived, but is it broadband? The text says so, but line 101 also contrasts itself by stating that only the narrow PAR band was studied.

Considering the generally cloudy weather in Arctic summer, the estimated albedo in the present study was white-sky albedo. Generally speaking, its value is slightly larger than the black-sky albedo. What we wished to express was that these optical properties were integrated in the PAR band. Related descriptions have been modified to reduce any ambiguity.

o ECMWF provides quite a few irradiance data, what is specifically the data source here?

The EAR5 monthly averaged surface downward shortwave radiation flux (https://cds.climate.copernicus.eu/cdsapp#!/dataset/reanalysis-era5-single-levels-monthly-means?tab=form) was employed in this study. More descriptions have been supplied in the data section (L132).

o The same for NSDIC sea ice concentration, did you use the old and outdated version 1 of NSDIC-0051? There is an update, which should be applied in all new studies: https://doi.org/10.5067/MPYG15WAA4WX

Yes, we have used the updated version in the revised manuscript.

The generalization part here is the major weakness of the manuscript and either requires considerable attention to improve it, of removal if improvements are not feasible.

1. Result figures 2-8 are not clear – for one, choosing to use red and green lines with the same markers renders the figures indistinguishable for color-blind readers. Also, the blue color chosen is very similar to the green, making it difficult to distinguish which is which even for me with normal color vision. Also, it would be clearer if the subplots such as Fig 3 had their own titles (e.g. "$V_a$" for Fig 3a), since each studied variable is given its own subplot anyway, so the legend does not have to repeat the variables, but would suffice to simply indicate the layer coloring.

We have revised all figures. The green lines were all redrawn in yellow.

The figure titles and legends have been modified accordingly to make them clearer.

2. The result figures display uncertainty ranges which are sometimes defined in caption and sometimes not. Consistency is needed. Also, the associated text only makes note of changes in e.g. IOPs but does not analyze the significance of the changes in relation to their uncertainty. For instance, is the 7.5% increase in Va between 2008 and 2016 a significant change when compared to the uncertainty range (standard deviation) of the samples? This treatment of uncertainty should be a part of all analyses done in the manuscript.

   The definition of uncertainty ranges in captions has been supplied (Fig 2-7, 10-11).

   We have supplied the treatment of uncertainties in the revised manuscript. The Analysis of variance (ANOVA) was supplied in all analyses. About 40% of analyses in Sections 3.1, 3.2, and 3.3 was revised. After the analysis of variance, we found that the variation in the top layer of ice between years was statistically insignificant. Meanwhile, the interannual variations in the interior layer were significant. These were also the main changes in the conclusion. Meanwhile, to reduce the impact of temporal variations in the ice cores on the ice microstructure, we have pre-processed the ice core data. The ice cores in each year have now been allocated different weights according to their sampling date. As a result, the deviation of melting days in single years was reduced by ~50.5%. The precise understanding of the effects of spatial variations was a big challenge because there was little quantitative knowledge about them. Instead, we have quantitatively analyzed their effects on interannual changes. Subsequently, the analyses and some conclusions have been reworded accordingly.

Minor comments (line):

ln 60: August is hardly the beginning of the melting season for Arctic sea ice, rather the opposite?

   According to the melt data from NASA, the ice cores in the present manuscript were all sampled in the late melting season, where the sea ice has been melting for a while (~59 days). We have rewritten this sentence in the revised manuscript (L71-72).

---

## Author Response (AR2)

Dear Handing editor Dr. Marie Dumont,

Thank you for handling our manuscript (ID: egusphere-2022-552). Based on comments from reviewers, we made further revision of our manuscript. Please see below point by point responses to the reviewers' comments (blue) The revised parts in the manuscript are marked in red.

Best regards,

MiaoYu and co-authors

**Anonymous referee #3**

The manuscript is well written though minor language edits are suggested.

R15. The first time the term Va is used it must be defined.

Defined accordingly (L15).

R17. Suggest to add over time or something instead of "in the study years" or even indicate which years are being addressed here.

Done (L17).

R60. detailly -> details

Corrected (L59).

R79. On R72 the time since melt onset was presented as 59 days, why wasn't this date used here instead of 30 days?

We have tested the effects of $D$ on weighted mean values. The results revealed that the weighted mean values and standard deviations were nearly constant when the $D$ was large enough. This can be seen from Figure A. 30 days was an appropriate value for most ice cores in the present study. We have added this information in the revised manuscript (L83-84).

[Figure]

Figure A. Effects of D on the weighted mean values and standard deviations. Taking the TL-Va of ice cores in 2008 as an example.

R147-149. Perhaps you can elaborate a bit about the possibilities of different locations and in how this affects the results?

The decrease of $V_a$ with the depth is a common feature in sea ice (e.g. Crabeck et al. 2016). This is controlled by the ice growth conditions (Some information was supplied in L90-93). So, it is expected that different locations can't affect this decreasing trend.

R155-156. The year of 2012 was one of the sea ice minimum record years, has this affected the results?

Yes, Figure 11a of the present manuscript also demonstrated that the number of melting days of ice cores was the largest in 2012 from 2008 to 2016. However, we didn't link the present result with this phenomenon directly, because the difference in the scale was so great. More ice cores would be needed to analyze the relationship between them.

Figure 2. the results are presented as a combination of FYI and MYI, would it be possible to see a separation between the different ice types instead in this graph?

As discussed in Section 4.2 (Fig. 11), we have shown the exact age of ice cores in each year but not sorted them into FYI and MYI classes. Then, we found that changing ice IOPs in each year was connected to the different combinations of FYI and MYI but

we didn't show information about FYI and MYI separately here. This was reported in Wang et al (2020) who used the same dataset.

R225-226. An ice thickness of 1m is rather thin sea ice. In the European Arctic I would perhaps pick an average of 2m instead. Could you provide some justification of the chosen ice thickness? Figure 6 seems to indicate that most ice is thicker than 1m also in the data used here.

In this section, we estimated the ice optical properties by setting a constant thickness reference (1m) to scale the ice microstructure relative between the surface and bottom. The actual mean thickness of all ice cores was 1.2 m. On the other hand, the reference thickness value doesn't affect the trends in Fig 6. We have added this information to the revised manuscript. (L232-234)

R306-307. The data used in the study from 2008 likely contain ice of the type observed at the start of this study. Will the timing (2008 vs 2021) of the ice be important to consider for this comparison?

First, as discussed in Section 4.2, we didn't find significant variations in the IOPs of the ice top layer (Fig. 10). So, it is expected that the timing of the ice sample can hardly affect the comparison. Second, the variations in IOPs in the present literature were still not clear, so it is difficult for us to consider the potential affecting factors. So, we pay more attention to the comparison of the IOPs range in Fig 9. We have added this statement to the revised manuscript. (L308-310).

R370-372. This sentence reads a bit strange the statistically significant trend is not what is being provided from ECMWF, consider rewriting it.

Revised accordingly (L378-380).

**Main references**

Crabeck, O. and Galley, R., et al., 2016. Imaging air volume fraction in sea ice using non-destructive X-ray tomography. The Cryosphere, 10 (3): 1125-1145. doi:10.5194/tc-10-1125-2016

Wang, Q. and Lu, P., et al., 2020. Physical Properties of Summer Sea Ice in the Pacific Sector of the Arctic During 2008–2018. Journal of Geophysical Research: Oceans, 125 (9). doi:10.1029/2020JC016371
* * *
**Anonymous referee #4**

I realize that this is a revised manuscript. In this manuscript, the authors have tried to address the issues on the variations of the inherent optical properties (IOPs) of summer Arctic ice and their effects on the radiation budget. By using the observations obtained from the CHINARE during the summers of 2008 to 2016, they found that the changing microstructure of interior ice from 2008 to 2016 has led to the significant decrease in the scattering coefficient. They also reveal that such variations of microstructure and IOPs play an important role in the radiation budget of Arctic sea ice.

The paper is very well written and very informative. I think it well fits the scope of the journal, however, I still have some comments and minor suggestions as follows.

Abstract

Line 15: The full name of Va should be given here.

Revised accordingly (L15).

Data and method

Lines 71~75: Confusing about this statement. Do the authors want to express that the variability of IOPs during entire summer is very small? If yes, how should I understand that when the surface melting in August is only 1/10 of that in July, the state of surface ice microstructure in August is like that in July? If no, can observations concentrated in August represent the entire summer? I think it is necessary to clarify the representativeness of observations rather than just use 'summer'.

According to the observation in MOSAiC, the surface scattering layer (SSL) of

sea ice re-formed within a couple of days after removal (Smith et al., 2022). Macfarlane et al. (2023) found that there are no clear temporal changes in the microstructure of surface ice in the entire July. Furthermore, other observations have revealed that the ice surface melt rate in August was lower than in July (e.g. Perovich, 2003). So, it is expected that the microstructure of the ice surface develops rapidly in the early melting season, and is similar in the mid- and late-melting seasons but not the entire melting season (or the entire summer). We have revised these descriptions to make them clear (L74, L76-79).

Can you provide more information on ice in each year (e.g., thickness, age) in this section? Because it is important to explain your results.

The general description of the thickness and age of ice cores has been added here (L70-71). More detailed information has been supplied in Fig 6 (ice thickness) and Fig 11 (ice age). So, we didn't show the details here repeatedly.

Section 2.3 Arctic-wide up-scaling: My main concern on this part of analysis is whether your observational results is suitable for all the Arctic sea-ice types. I think more details should be concerned when you do up-scaling. For example, setting different values for the first-year and multi-year ice.

The different values for the first-year and multi-year ice have been considered in this section. As shown in Fig. 11, the age of ice cores in each year was different. That's to say, the changing ice IOPs resulted from different combinations of FYI and MYI in each year. L424-441 has in detail discussed the relationship between ice age, ice IOP, and ice microstructure.

Results

Line 203: 'there are' −> 'there were'.

Revised accordingly (L209).

Lines 209~210: 'Furthermore, the developments of $\kappa B$ in the three layers are similar (Figure 5b)'. Can you give correlation coefficients to support this statement?

Three fitting trends have been added here (L216).

Discussion

Figure 9: It is interesting that other studies present a noticeable smaller variation in σ for IL compared with your results? Can you explain this difference?

We have discussed the possible reason for this difference in Section 4.1 in detail. First, there isn't a unified method to get the ice IOPs to date. The result of IL from Light et al. (2015) and Frantz et al. (2019) were obtained by the same method, and they are smaller than the present result. Results from Mobley et al. (1998), Grenfell et al. (2006), and Perron et al. (2021) agree with our results. The method of these four studies was not the same. So, it was expected that the differences in the IL's $\sigma$ partly resulted from the different methods used in the myriad studies (See details in L317-325).

Another possible reason comes from the brine loss during measurement and segmenting. Thus, our $V_a$ values of the IL may be greater than the values derived from nondestructive methods. Taking the mean $V_a$ and $V_b$ of all ice cores as an example, these uncertainties overestimated the $\sigma$ of the IL by 78 m$^{-1}$ at most. This information has been supplied in detail (L326-333).

**Main references**

Smith, M. M. and Light, B., et al., 2022. Sensitivity of the Arctic Sea Ice Cover to the Summer Surface Scattering Layer. Geophysical Research Letters, 49 (9): e2022GL098349. doi:10.1029/2022GL098349

Perovich, D. K., 2003. Thin and thinner: Sea ice mass balance measurements during SHEBA. Journal of Geophysical Research, 108 (C3). doi:10.1029/2001JC001079

Macfarlane, A. R. and Dadic, R., et al., 2023. Evolution of the microstructure and reflectance of the surface scattering layer on melting, level Arctic sea ice. Elementa: Science of the Anthropocene, 11 (1). doi:10.1525/elementa.2022.00103